# Influence of microbiota-associated metabolic reprogramming on clinical outcome in patients with melanoma from the randomized adjuvant dendritic cell-based MIND-DC trial

Tumor immunosurveillance plays a major role in melanoma, prompting the development of immunotherapy strategies. The gut microbiota composition, influencing peripheral and tumoral immune tonus, earned its credentials among predictors of survival in melanoma. The MIND-DC phase III trial (NCT02993315) randomized (2:1 ratio) 148 patients with stage IIIB/C melanoma to adjuvant treatment with autologous natural dendritic cell (nDC) or placebo (PL). Overall, 144 patients collected serum and stool samples before and after 2 bimonthly injections to perform metabolomics (MB) and metagenomics (MG) as prespecified exploratory analysis. Clinical outcomes are reported separately. Here we show that different microbes were associated with prognosis, with the health-related *Faecalibacterium prausnitzii* standing out as the main beneficial taxon for no recurrence at 2 years ($p = 0.008$ at baseline, nDC arm). Therapy coincided with major MB perturbations (acylcarnitines, carboxylic and fatty acids). Despite randomization, nDC arm exhibited MG and MB bias at baseline: relative under-representation of *F. prausnitzii*, and perturbations of primary biliary acids (BA). *F. prausnitzii* anticorrelated with BA, medium- and long-chain acylcarnitines. Combined, these MG and MB biomarkers markedly determined prognosis. Altogether, the host-microbial interaction may play a role in localized melanoma. We value systematic MG and MB profiling in randomized trials to avoid baseline differences attributed to host-microbe interactions.

The rise in incidence of melanoma worldwide has led to an increasing number of patients with regional positive lymph nodes (stage III) being diagnosed each year, especially in the western world[1]. Over the last decade, there has been tremendous progress in the clinical management of stage III melanoma with the advent of adjuvant and neoadjuvant immune checkpoint inhibitors (ICI). Before the era of (neo) adjuvant ICI in localized melanoma, patients with operable clinically positive nodes systematically underwent full lymphadenectomy of the involved sites[2,3]. Surgery alone is insufficient to achieve a cure in most patients with high-risk stage III melanoma. Thus, systemic adjuvant therapy has been investigated over the last decades in patients with high-risk melanoma. The development of effective adjuvant therapies

e-mail: laurence.zitvogel@gustaveroussy.fr

for patients with high-risk melanoma has included ipilimumab (an anti-CTLA-4 antibody), pembrolizumab and nivolumab (both monoclonal antibodies against programmed death 1 [PD-1]), and combination of BRAF and MEK inhibitors for patients whose tumors harbor a BRAF mutation[1,4–12], leading to US Food and Drug Administration approvals. More recently, pembrolizumab and the combination of anti-CTLA-4 and PD-1 antibodies showed benefit in the neoadjuvant setting in patients with high-risk node-positive melanoma in early phase clinical trials[13–16].

Prior to the modern era of ICI, vaccination involving dendritic cells (DC) has been developed owing to the special properties of these cells in coordinating innate and adaptive immune responses. The aim of DC immunization was to induce tumor-specific effector T cells that can exhibit a tumoricidal activity in a tumor antigen-specific manner through induction of a protective immunological memory to cancer antigens. Across the world, many investigators showed that DC-based vaccines were safe and induced the expansion of circulating CD4+ T cells and CD8+ T cells that were tumor antigen-specific. Objective clinical responses have been observed in 5–8% of patients[17–21]. Long term follow-up of DC-vaccinated patients with metastatic melanoma (MM) reported up to 19% survival rates at 11 years, comparable to ipilimumab-treated patients. Survival significantly correlated with intense reactivities at the dermal injection site, and with eosinophilia[22]. In the past years, DC-based immunotherapy was performed in patients with stage IV HLA-A2.1 positive melanoma using intravenous, intradermal, and intranodal routes of administration of mature DC loaded with tumor-associated antigens (TAA) such as tyrosinase and gp100, and keyhole limpet hemocyanin (KLH) as a control antigen. All vaccinated patients showed a pronounced proliferative T cell or humoral response against KLH. TAA-specific T cell reactivities were monitored in post-treatment delayed-type hypersensitivity (DTH) skin biopsies by tetramer staining and functional analysis. Patients harboring peptide-specific T cell immunity exhibited the best clinical response, with eventually complete responses in a minority of patients with MM while rIL-2 or modified TAA did not further increase vaccine efficacy[23,24]. A direct correlation between DC-induced tumor-specific T cells detected in DTH skin biopsies and a favorable clinical outcome was observed in patients with MM[25,26].

The "Melanoma Patients Immunized With Natural DenDritic Cells (MIND-DC)" trial was a randomized phase III clinical trial testing adjuvant natural DC (nDC) therapy in high-risk stage III melanoma. The trial showed that adjuvant nDC treatment generated specific immune responses but did not translate into survival benefit[27]. Here, we investigated whether nDC loaded with TAA ex vivo could modulate fecal metagenomics (MG) and serum metabolomics (MB) profiles that might in turn, influence clinical outcomes. Indeed, shotgun MG-based taxonomic composition of feces at baseline was associated with objective response rates in patients with stage IV melanoma treated with anti-PD-1 alone or combined to anti-CTLA-4 in several independent cohorts[28–31]. In a meta-analysis incorporating new cohorts, McCulloch et al. confirmed that baseline microbiota composition was associated with 1-year progression-free survival. Bacteria associated with favorable response during ICI belonged to *Lachnospiraceae* and *Bifidobacteriaceae* families including *Ruminococcus* spp, *Mediterraneibacter* spp., and *Blautia* spp.[32].

Here, we show that a relative deficiency in the primary biliary acid (BA) cholic acid or *F. prausnitzii*, or high levels of fatty acids (FA) and acylcarnitines are associated with reduced recurrence-free survival (RFS), especially in the nDC treatment arm. Moreover, the relative abundance of beneficial *F. prausnitzii* in stool anti-correlate with serum BA and FA. Therefore, we hypothesize that the pharmacodynamic effects of the nDC might have been influenced by the host-microbiome dialogue in patients harboring a deviated lipid metabolism (including carboxylic acids, BA and acylcarnitines) and gut microbiota.

## Results

### Metagenomics-based profiles at baseline (T1) and at 4 weeks (T2) are associated with 2 year-RFS (2Y-RFS)

The MIND-DC trial (NCT02993315) randomized 148 eligible patients with resected stage IIIB and IIIC melanoma between 2018 and 2021, of which 144 patients were included in this translational cohort ($n = 95$ in the nDC arm, versus $n = 49$ in the placebo (PL) arm) (Supplementary Table 1). Five patients (all in the nDC arm) never received any injection, three being related to tumor recurrence. One was censored (because of a follow up inferior to 1 month). Primary and secondary clinical outcomes are reported separately[27]. Shotgun MG sequencing of stools at baseline (pre-therapy, T1) and after 4 weeks of therapy start (at the day of the first and the third biweekly intranodal injections of nDC or PL, T2) was undertaken, and the presence and abundance of species-level genome bins (SGBs) was estimated[33]. Here, we describe a longitudinal follow up of the shotgun MG-based stool composition in 88 and 85 patients with stage IIIB/C melanoma at T1 and at T2, respectively (Supplementary Table 2). Firstly, we correlated the microbiota composition of the T1 (Fig. 1A) and T2 (Fig. 1B) feces samples with 2Y-RFS. While the alpha-diversity was not associated with 2Y-RFS ($p = 0.2$ at T1 and $p = 0.1$ at T2, Shannon index), the beta-diversity significantly differed between those patients prone to stay cancer-free (no recurrence at 2 years, 2Y-noR) versus those who will experience a 2-year recurrence (2Y-R) at both time points ($p < 0.01$ at T1 and T2, Bray-Curtis dissimilarity). We focused on differentially abundant SGBs according to 2Y-RFS in the whole population and found a relative overrepresentation of *Faecalibacterium prausnitzii* (whether considering SGB15318 or SGB15322) in patients with favorable prognosis at T1 (Fig. 1A, Supplementary Data 1) and T2 (Fig. 1B, Supplementary Data 1).

By splitting the whole cohort into 4 groups according to treatment arm and outcomes, we obtained small sample sizes (Supplementary Table 2). This prevented any significant associations to pass the FDR correction for the MG analysis (all Q-values $\geq 0.2$). Linear model coefficients (MaAsLin2, coefficient) for microbial SGBs that were found associated either after arcsine square root (arcsin-sqrt) transformation (AST) or centered-log-ratio (CLR) transformation with 2Y-R with $p < 0.05$ at T1 and T2 are detailed in Supplementary Fig. 1A–D and Supplementary Data 2. In brief, *F. prausnitzii* SGB15318 was relatively over-represented in 2Y-noR in the nDC arm at T1 and T2 (Supplementary Fig. 1B–D, $p < 0.05$). Of note, *Ruminococcaceae* SGB14899, *Streptococcus salivarius* SGB8007, and *Streptococcus parasanguinis* SGB8071 followed the same behavior only in the nDC arm (Supplementary Fig. 1B,D, $p < 0.05$). In addition, the prevalence and relative abundances of distinct SGBs of *F. prausnitzii* tended to be reduced in the nDC arm compared with the PL arm (Supplementary Fig. 2A, B). The MG biomarker evolution between the two timepoints was assessed using two methods: paired-Wilcoxon test (Wilcoxon signed-rank test) (Supplementary Fig. 3) and linear regression adjusted for the clinical features treatment arm, age, gender, tumor stage (IIIB vs IIIC), Eastern Cooperative Oncology Group performance status (ECOG-PS), and body mass index (BMI) (Supplementary Fig. 4). Of note, the nDC treatment barely impacted on the shift of the MG taxonomic composition, except for a few taxa (*Blautia sp MSK 20 85* SGB4828, $p = 0.021$, and *Ruminococcus torques* SGB4608, $p = 0.024$) (Supplementary Fig. 4). Of note, these dynamics did not pass the FDR correction. We may impute these weak associations to the small sample size (limiting the power of the analysis) and to the lack of clear effect of the treatment arm in this negative trial.

We concluded that gut microbiota composition was associated, albeit weakly, with the prognosis of stage III melanoma, with *F. prausnitzii* (SGB15318 and SGB15322), referenced for its homeostatic[34,35] and antitumor properties in MM[30,36], as the most prominent MG species (MGS) found at T1 and T2 in this cohort.

## Serum metabolic changes following the first cycle of immunization

In parallel to MG, the mass spectrometry-based serum MB was serially assessed at T1 and T2 in 95 patients in the nDC arm and 49 patients in the PL arm. The Volcano plot and principal component analysis (PCA) revealed differences between the two time points overruling those observed between the treatment groups (Fig. 2A, B, Supplementary Fig. 5A, Supplementary Data 4). In particular, the lipid metabolism

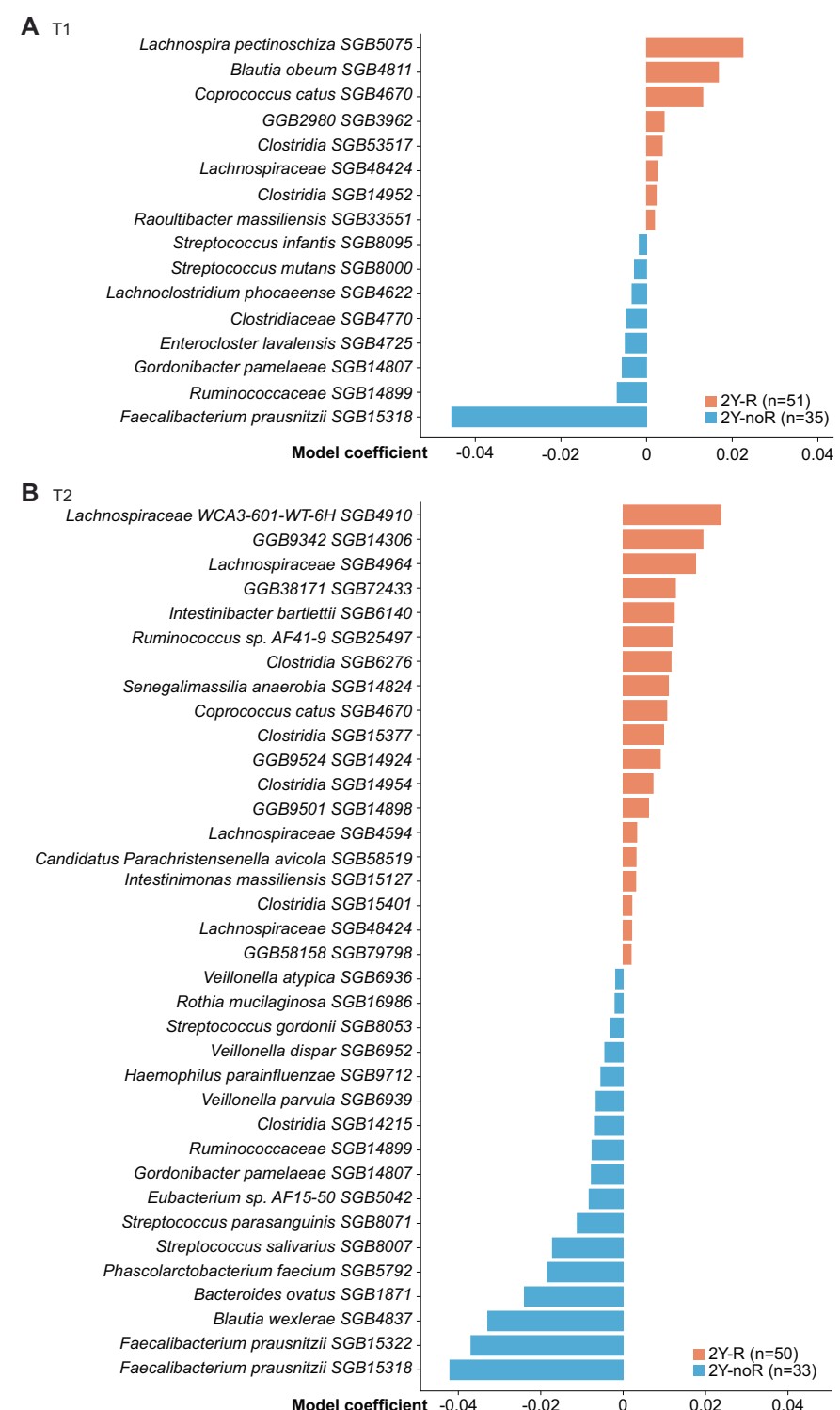

**Fig. 1 | Metagenomics-based profiles show taxonomic signatures associated with recurrence at 2 years (2Y-R) in two different time points.** A, B Linear model coefficients for microbial SGBs associated either with 2y-R or 2y-noR, corrected for age, gender and treatment arm at baseline (**A**, T1 $n = 86$) or after 4 weeks of therapy start (**B**, T2 $n = 83$). Positive values indicate species-level genome bin (SGB) association with 2Y-R (orange), while negative values indicate a positive association for the corresponding SGB with 2Y-noR (blue). Only associations with $p \leq 0.05$ are reported since no association has Benjamini-Hochberg Q < 0.2. Refer to Supplementary Data 1 for linear model coefficients (MaAsLin2, coefficient) for microbial SGBs after arcsine square root (arcsin-sqrt) transformation (AST). Source data are provided as a Source Data file.

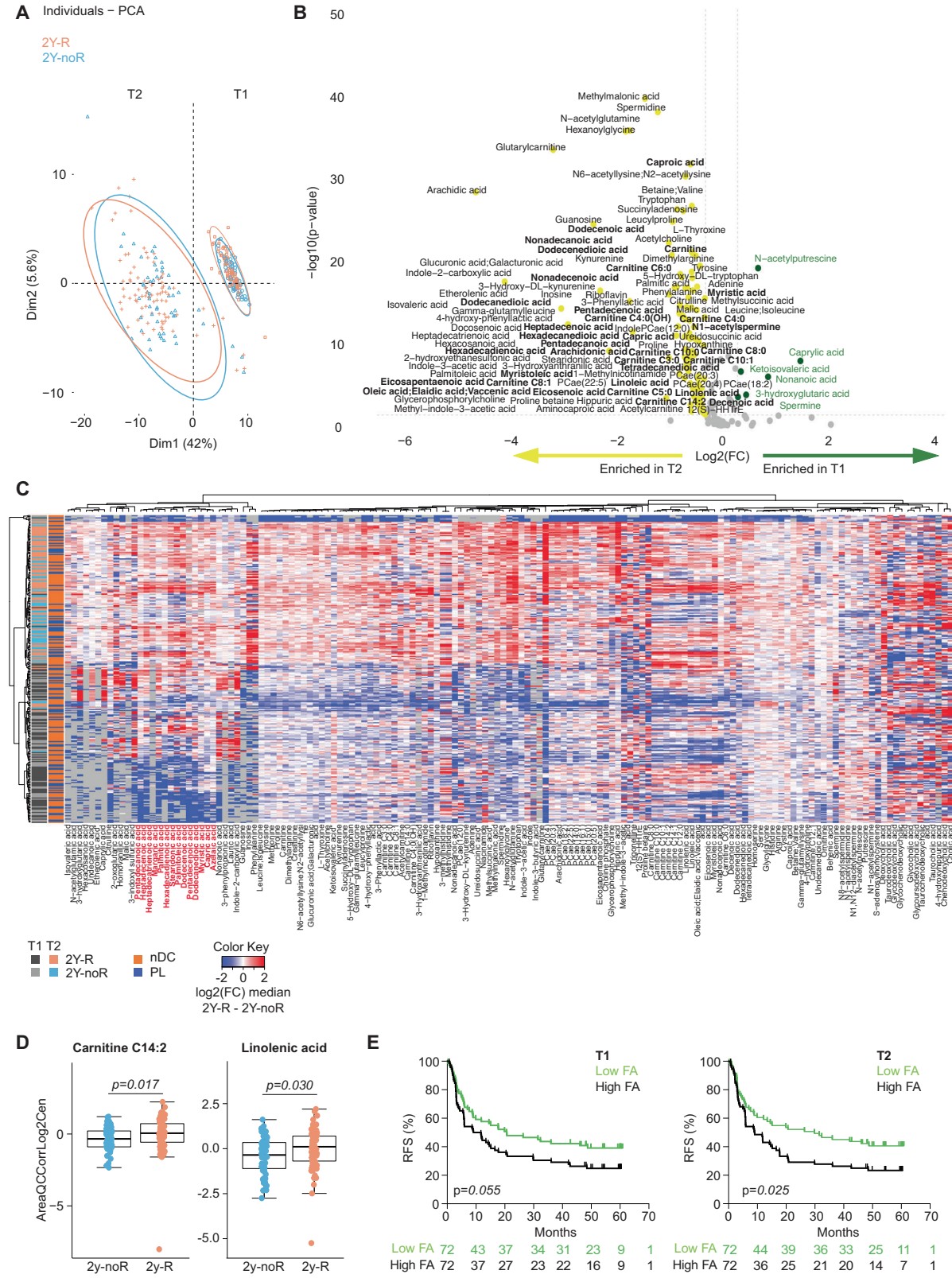

became perturbed at T2, with the accumulation of very long and long chain saturated FA (such as the carboxylic acids docosanoid acid (also called "behenic" acid), nonadecanoic acid, tetradecanedioic acid, arachidonic acid, hexacosanoic acid, palmitic acid), and mono- or poly-unsaturated FA (such as 10-nonadecenoic acid, oleic acid, linoleic acid, linolenic acid, stearidonic acid, docosenoic acid) as well as acylcarnitines (glutarylcarnitine, carnitine C4:0, C4:0 (OH), C6:0, C8:1, C10:1,

C12:0, C14:2) (Fig. 2B, $p < 0.05$, Mann–Whitney). Besides the shift in circulating FA and in the acylcarnitine shuttle found at T2, we observed perturbations of polyamine biosynthesis and acetylation between T1 and T2 (N-acetylputrescine, spermidine, N1-acetylspermine) (Fig. 2B, $p < 0.05$, Mann–Whitney). In addition, branched-chain amino acids (BCAAs) (valine/leucine/isoleucine), and γ-glutamyl dipeptides involved in inflammation, oxidative stress, and glucose regulation

**Fig. 2 | Longitudinal metabolic patterns showing a shift in lipid metabolism associated with recurrence at 2 years (2Y-R). A** Principal component analysis (PCA) plot representing the distribution of serum metabolomics (MB) overtime and according to 2y-R (orange, $n = 83$) versus no recurrence at 2 years (2y-noR, blue, $n = 56$). Circle: 2y-noR at baseline (T1); triangle: 2y-noR after 4 weeks of therapy start (T2); Square: 2y-R at T1; Crosses: 2y-R at T2. **B** Volcano-plots based on MB showing differences ($p < 0.05$, two-sided Mann−Whitney U-test with no adjustment) between T1 (green dots, $n = 143$) and T2 (yellow dots, $n = 143$) with a cut-off in the T2/T1 fold change (FC) ≥ 0.3. Metabolites with T2/T1 FC ≥ 0.3 with a $p < 0.05$ were colored green. X-axis: log2 fold change of metabolites; Y-axis: fold change of −log10. **C** Hierarchical clustering of MB according to 2y-noR ($n = 56$) versus 2y-R ($n = 83$) at T1 and T2 and treatment arm. Dark gray: 2y-R at T1; Light orange: 2y-R at T2; Light gray: 2y-noR at T1; Light blue: 2y-noR at T2. Dark blue: placebo (PL) arm ($n = 47$); Dark orange: natural Dendritic Cell (nDC) arm ($n = 92$). Targeted MB computed as normalized areas of identified metabolites. Heatmap illustrating the changes in metabolite abundances according to the median of each metabolite in the two subgroups of opposite prognosis, highlighting the fatty acids (FA). Rows are samples, columns are metabolites. Heatmap data are log2 normalized and centered around the average abundance computed from all the samples for each metabolite. Red/blue colors are ion signal higher/lower than average and gray are missing values. Samples are sorted following biological conditions and metabolites clustered following the ward.D2 algorithm, with euclidean distance. **D** Relative abundance of Carnitine C14:2 (left panel) and Linolenic acid (right panel) in 2y-noR (blue: $n = 56$) and 2y-R (orange: $n = 83$ at T1). Boxplots indicates the interquartile range Q1 to Q3 with Q2 (median) in the center. The range of outliers is depicted by whiskers. The $p$ value are related to the group comparison using the two-sided Mann−Whitney U-test with no adjustment. **E** Recurrence-free survival (RFS) analysis using the Kaplan−Meier estimator (Log-Rank (Mantel Cox) test) to assess low FA versus high FA (calculated based on the sum of relative abundances of 13 most significant FA or carboxylic acids) at T1 (left panel) and at T2 (right panel).

were significantly increased at T2 (Fig. 2B, $p < 0.05$, Mann−Whitney)[37]. These metabolic shifts could not be explained by clinical events between the two time points since few patients experienced flu-like symptoms (19%) or started new medications (12%) between T1 and T2 (Supplementary Table 3).

To assess the clinical relevance of these early changes, we investigated which metabolic profiles were associated with 2Y-RFS using an unsupervised hierarchical clustering across all metabolites (Fig. 2C). We concluded that the above-mentioned medium and long chain-acylcarnitines and FA linolenic acid (Fig. 2D, $p < 0.05$ Mann-Whitney), acetylated polyamines (Supplementary Fig. 5B, $p < 0.05$ Mann−Whitney), as well as BCAAs and γ-glutamyl dipeptides (Supplementary Fig. 5C, $p < 0.05$ Mann−Whitney) were negatively associated with 2Y-RFS at T1 in univariate analysis. In contrast, serum levels of ornithine, a precursor of spermidine, were higher in patients with favorable prognosis (Supplementary Fig. 5D, $p < 0.05$ Mann−Whitney) in univariate analysis.

The significance of FA in the temporal metabolic shift and clinical recurrence prompted us to evaluate the RFS according to the median of the sum of the 13 most relevant FA abundances. Levels above the FA median at T2 ($p = 0.025$, right) and to a lesser extent at T1 ($p = 0.055$, left) were associated with shorter RFS (Fig. 2E).

Attempting to ascribe these lipid profiles to the gut microbiome, we used the MG data, and annotated the organism-specific gene hits according to the UniRef90 (UR90)[38]. Based on these annotations, metabolic pathways for each sample were then defined using the MetaCyc hierarchy of pathway classifications[39]. The Linear model-based analysis contrasting hits separating patients who recurred of their melanoma from the ones who did not recur showed that the FA beta-oxidation and biosynthesis pathways at T1 were associated with overall recurrence (Supplementary Fig. 6A).

Hence, these findings suggest that time and/or therapy have perturbed a delicate and preexisting disbalance of lipid synthesis or degradation, at the level of the mono- and poly-unsaturated and saturated fatty and carboxylic acid metabolism, coinciding with melanoma recurrence.

## Taxonomic and metabolomic differences between treatment arms at randomization

Our findings indicate that the gut taxonomic composition at T1 and the serum metabolic profile shift after only 2 injections of nDC treatment were associated with the 2Y-RFS. Although we did not anticipate any significant difference in clinical characteristics between the two treatment arms at baseline given the process of randomization (Supplementary Table 1), we took a deeper dive into potential pre-existing differences in the fecal microbial ecosystem defined by shotgun MG-based sequencing. Strikingly, while the stool compositional diversity did not differ between the two arms ($p = 0.43$, Shannon index), the beta-diversity[40] of the taxa present in feces from stage III melanoma randomized to PL was significantly different from that of individuals about to receive nDC ($p = 0.02$, Bray-Curtis dissimilarity). Patients randomized in the PL arm exhibited a relative over-representativity of health-related and immunogenic MGS *F. prausnitzii* SGB15322, *Blautia massiliensis* SGB4826, and *Dorea formicigenerans* SGB4575 compared with the nDC arm[32,41] (Fig. 3A, Supplementary Data 3). Indeed, the nDC treatment arm tended to harbor higher proportions of individuals lacking *F. prausnitzii* spp. (Fig. 3B) and more specifically distinct SGBs of *F. prausnitzii* or bearing lower relative abundance of these SGBs compared to the PL arm (Supplementary Fig. 2B). The same observation could be made comparing stage IV melanoma from publicly available databases with healthy volunteers (HV)[42] (Supplementary Fig. 2A).

When we analyzed the effect of nDC (versus PL) on the RFS according to the relative abundance of several MGS associated with survival, using Kaplan−Meier (KM) analysis, we observed that the presence of higher relative abundances of commensals associated with recurrence aligned in Fig. 1B and Supplementary Fig. 1B (such as *Gemmiger formicilis*, $p = 0.270$, and *L. pectinoschiza*, $p = 0.009$, Log-Rank (Mantel Cox) test) (Fig. 3C) and the absence or low abundances of bacteria associated with good prognosis (such as *F. prausnitzii*, $p = 0.097$, Log-Rank (Mantel Cox) test) (Fig. 3D) were associated with a poor prognosis in the nDC arm. Of note, many other commensals appeared irrelevant to predict RFS with or without nDC.

Next, to evaluate if this MG bias also translated into different MB patterns between the two treatment arms at T1, we re-analyzed the mass-spectrometry-based MB profiles of PL and DC arms using a supervised hierarchical clustering (Fig. 4A, Supplementary Data 5). Strikingly, there was a highly significant bias in the cluster corresponding to primary conjugated biliary acids at T1 (Fig. 4B, $p < 0.05$ Mann−Whitney) that was confirmed at T2 (Fig. 4C, $p = 0.002$ and $p < 0.001$ Mann−Whitney) in the nDC arm. Another cluster composed of mono-unsaturated and long chain FA separated nDC from PL groups at T1 (Fig. 4A). To identify whether these perturbations in BA and FA translated into clinically relevant differences, we compared the most significant concentration differences between 2Y-noR and 2Y-R in PL and nDC groups for each of these metabolites and performed RFS KM curves. High levels of acylcarnitines C12:0 and C14:1, as well as linolenic acid were markedly associated with recurrence in the nDC arm only (Fig. 4D, $p < 0.05$ Mann−Whitney) in contrast to the primary BA cholic acid, that was associated with prolonged RFS (Fig. 4D, $p < 0.05$, Mann−Whitney, and Fig. 4E, $p = 0.057$, Log-Rank Mantel Cox test), but not in the PL arm (Supplementary Fig. 6B).

Hence, despite randomization, patients assigned to the two treatment arms appeared to differ as to the relative representativity of *F. prausnitzii* SGBs and primary BA composition.

## Dynamic and integrative pathways

Using the XGBoost algorithm, coupled to a model explainer based on SHapley Additive exPlanations (SHAP) values for model interpretability[43], we corroborated that patients assigned to nDC arm differed from patients in the PL arm as to primary BA and polyamines (Supplementary Fig. 7).

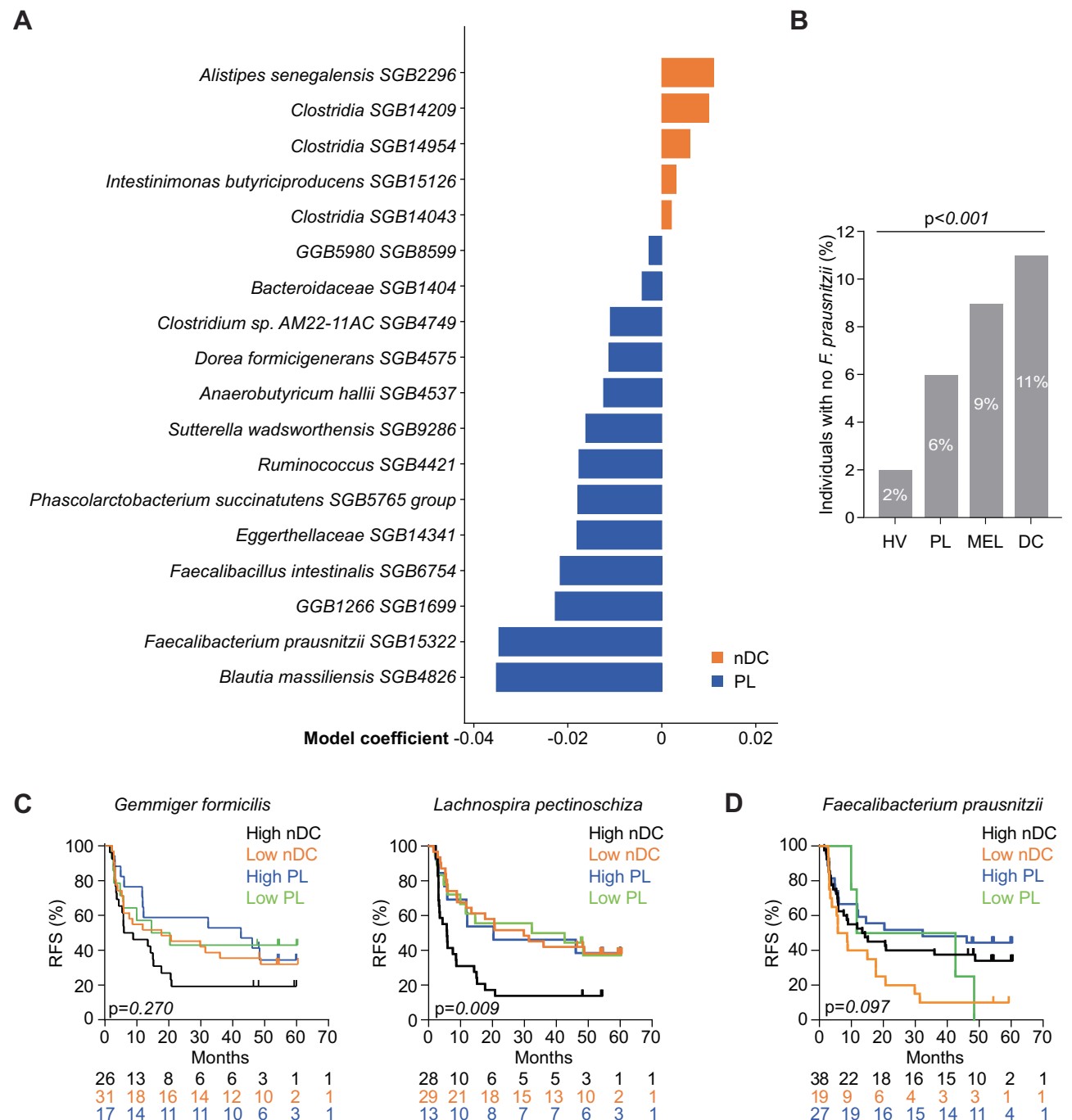

**Fig. 3 | Significant differences in the microbiota taxonomic profiles at randomization.** A Linear model coefficients for microbial SGBs differentially abundant either in the natural dendritic cell (nDC, $n = 56$) or placebo (PL, $n = 31$) arms, corrected for age and gender. Positive values indicate species-level genome bin (SGB) association with nDC (orange), while negative values indicate a positive association for the corresponding SGB with PL (blue). Only associations with $p ≤ 0.05$ are reported since no association has Benjamini-Hochberg $Q < 0.2$. Refer to Supplementary Data 3 for linear model coefficients (MaAsLin2, coefficient) for microbial SGBs after arcsine square root (arcsin-sqrt) transformation (AST). Source data are provided as a Source Data file. **B** Prevalence of *Faecalibacterium prausnitzii*, i.e., proportion of individuals with its absence between healthy volunteers (HV,

$n = 5345$), patients into PL arm ($n = 31$), all patients with melanoma into MIND-DC trial (MEL, $n = 88$) and nDC arm ($n = 57$). The $p$ values are related to the group comparison using the Chi-square test ($p < 0.0001$). Recurrence-free survival (RFS) analysis using the Kaplan–Meier estimator (Log-Rank (Mantel Cox) test) to assess the predictive value of *Gemmiger formicilis* (**C**, left panel) and *Lachnospira pectinoschiza* (**C**, right panel) and *F. prausnitzii* (**D**) using relative abundances at T1. The two groups of patients were defined by reference values of relative abundances (MetaPhlAn 4) from publicly available HV cohort: high if ≥ median and low if <median of the metagenomic species' relative abundance from HV. The numbers per group are depicted under the plots.

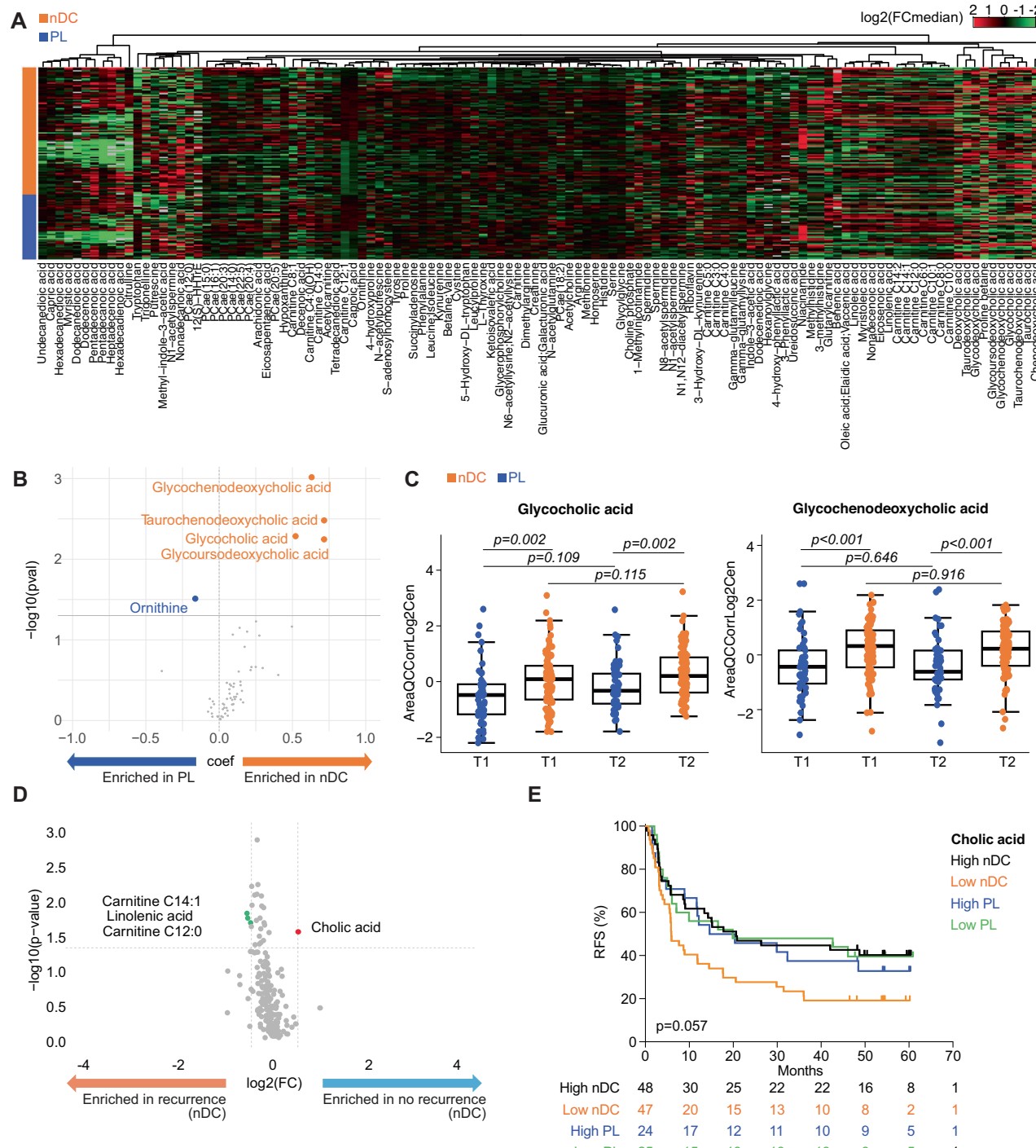

**Fig. 4 | Significant differences in metabolomics (MB) profiles at randomization.** **A** Hierarchical clustering of metabolites according to the randomization arm: placebo (PL, $n = 49$) versus natural Dendritic Cell (nDC, $n = 94$) at baseline (T1)). Targeted MB data on serum samples computed as normalized areas of identified metabolites. Heatmap illustrating the changes in metabolite abundances according to the median of each metabolite in the two arms. Rows are samples, columns are metabolites. Heatmap data are log2 normalized and centered around the average abundance computed from all the samples for each metabolite. Red/green colors are ion signal higher/lower than average and missing values are displayed as gray. Samples and metabolites are cluterized following the ward.D2 algorithm, with euclidean distance. **B** Volcano-plots based on metabolomic data showing significant ($p < 0.05$) differences between PL (blue dots, $n = 49$) and nDC (orange dots, $n = 94$) with a cut-off in the fold change (FC (DC/Placebo)) ≥ 0.5. X-axis: log2 fold change of metabolites; Y-axis: fold change of −log10. The $p$ value determined by the two-sided Mann−Whitney $U$-test with no adjustment. **C** Relative abundances of key

metabolites (glycoconjugated primary bile acids) from significant perturbations detected in (**B**). between nDC (orange, $n = 94$) and PL (blue, $n = 49$) arms at T1 and T2. Boxplots indicates the interquartile range Q1 to Q3 with Q2 (median) in the center. The range of outliers is depicted by whiskers. The $p$ value are related to the group comparison using the two-sided Mann−Whitney $U$-test with no adjustment. The exact $p$ values are reported in Supplementary Data 5. **D** Volcano-plots based on metabolite significant differences in the nDC arm at T1 ($n = 95$) associated with recurrence (R) (orange, left) versus no recurrence (noR) (blue, right). Metabolites with FC (R-nDC/noR-nDC)) ≥ 0.5 and $p < 0.05$ were colored in red dots, while those with FC (R-nDC /noR-DC) < 0.5 and $p < 0.05$ were colored in green dots. X-axis: log2 fold change of metabolites; Y-axis: fold change of −log10 $P$ value determined by the two-sided Mann−Whitney $U$-test with no adjustment. **E** Recurrence-free survival (RFS) analysis using the Kaplan−Meier estimator (Log-Rank (Mantel Cox) test) to assess the predictive value of Cholic acid abundance at T1.

Next, we re-analyzed the clinical relevance of all biological and clinical features to reduce dimension and allow prediction of 2Y-R taking into account their interactions and their slope of evolution overtime. We corroborated that primary BA, FA, polyamines and acylcarnitines as well as a few pathways meaningful in other studies including tryptophan metabolism, vitamin B3 (trigonelline, 1-methyl-nicotinamide) were predictors of 2Y-RFS at T1 (Fig. 5A, AUC = 0.74; 95% CI: 0.74−0.77). We confirmed the prognostic and independent impact of *F. prausnitzii* SGB15318 at T1 (Fig. 5A). Next, the metabolite evolution between the two timepoints (T2-T1/T1) was added to the baseline value (T1) to identify if biomarker drifts could improve the prediction of the 2Y-RFS and if treatment arms influenced this evolution (Supplementary Fig. 8A–B). The SHAP analysis indicated that the increase of FA (oleic acid) and acetylated polyamines and the decrease of ornithine (upstream of the polyamine pathway) and primary BA (glyco-cholic acid, chenodeoxycholic acid) were associated with an increased risk of 2Y-R (Supplementary Fig. 8A, T2-T1/T1: AUC = 0.72 (95%CI: 0.66−0.78). Even if the trajectory of BA and polyamines was associated with the prognosis of stage III melanoma, we could not conclude that nDC significantly alter their levels overtime (Supplementary Fig. 8B, $p = 0.088$ and $p = 0.064$ respectively). We applied the same analysis integrating MG and clinical features at T1, but obtaining better prediction at T2 for 2Y-RFS (Fig. 5B, AUC = 0.79 (95%CI: 0.71; 0.79)). The multi-omics integrative model depicted in the circosplot highlights positive correlations between many of these parameters of dismal prognosis and anticorrelations between tumor stage and factors associated with better prognosis such as cholic acid, polyamines and *Ruminococcaceae bacterium* SGB14899 at T1 (Fig. 5C).

We next performed Pearson correlations (Supplementary Fig. 9A–D) between fecal relative abundance of *F. prausnitzii* (SGB15318 and SGB15322) and serum MB at T1 and T2 in each treatment arm. The significant metabolites correlating with *F. prausnitzii* SGBs were analyzed by metabolic pathway-enrichment analysis performed in MetaboAnalyst using a KEGG database[44] (Supplementary Fig. 9E). Both *F. prausnitzii* SGBs markedly anticorrelated with many acylcarnitines, most specifically with medium chain (C6:0, C8:0, C10:0) and long chain saturated (C14:1) acylcarnitines, as well as many fatty and carboxylic acids (linoleic and linolenic acids) at T1 and/or T2, mainly in the nDC arm (Supplementary Fig. 9A–D). The 13 serum FA abundance were clinically relevant after 2 injections of treatment. Indeed, levels above the median at T2 were associated with earlier recurrence in the nDC arm ($p = 0.013$, Log-Rank (Mantel Cox) test) but not in the PL arm ($p = 0.640$, Log-Rank (Mantel Cox) test) (Fig. 6A). Low levels of acylcarnitines, i.e., lauroyl-L-carnitine (C12:0) or myristoylcarnitine (C14:1) at T1 were associated with prolonged RFS in the PL arm (Fig. 6B left, $p = 0.049$ and Fig. 6B right, $p = 0.099$, Log-Rank (Mantel Cox) test), with no significant effect in the nDC arm (Fig. 6B). In fact, taking into account both *F. prausnitzii* relative abundance and acylcarnitine estimates, we could identify a subgroup of nDC-treated patients with improved RFS. Patients in the nDC arm harboring high relative abundance of *F. prausnitzii* (*F. prausnitzii*^high) and low levels of acylcarnitines (either carnitine C12:0^low or C14:1^low) representing up to 37% (22/60) of the nDC arm exhibited a prolonged RFS (Fig. 6C left, $p = 0.011$ and Fig. 6C right, $p = 0.034$, Log-Rank (Mantel Cox) test). The survival advantage of this subgroup of individuals based on these two biomarkers could be generalized to all patients (Supplementary Fig. 10A, B). Supplementary Fig. 10C, D depicts the effects of distinct SGBs of *F. prausnitzii* (SGB15316 and SGB15318 being the most impactful ones).

We conclude that miscellaneous serum soluble biomarkers independent of clinical parameters impacted the survival of this cohort of stage III melanoma, that may not be directly inferred to the nDC treatment, including *F. prausnitzii* strains, FA, acylcarnitines, BA and polyamines.

## Discussion

This ancillary prospective study attempted to infer a prognostic impact of serum biomarkers analyzed at two time points in the treatment efficacy of nDC vaccination. Due to treatment failure and to relatively small subgroup of patients, it aims at drawing new hypothesis on how systemic pathological deviations in the host-microbiome interaction may be clinically relevant. We conclude that, despite the randomization process, a bias was inadvertently introduced in the MIND-DC trial with the nDC treatment arm skewed at baseline towards a relative under-representation of health-associated commensals, dominated by but not limited to distinct SGBs of *F. prausnitzii*[32,45]. This bias also translated into a relative increase of the conjugated primary BA in the nDC arm (compared with the PL arm), likely at the expense of the unconjugated BA (namely cholic acid harboring a favorable prognostic value). Interestingly, the *F. prausnitzii* SGBs anticorrelated with fatty and carboxylic acids, as well as acylcarnitines, especially in the nDC arm. Last but not least, both *F. prausnitzii* and FA metabolites (acylcarnitines) cooperated to predict RFS in the whole cohort, identifying a subset of patients with better prognosis in the nDC arm.

This unexpected randomization bias was not due to artefactual deviations in serum sample storage or freezing troubleshooting with center of enrollment (Zwolle versus Nijmegen) effects or co-medications to the best of our appreciation. *F. prausnitzii* is one of the main members of the *Faecalibacterium* genus within the Firmicutes phylum, described as a health-related gut homeostatic bacterium[46]. Defects in the relative abundance of *F. prausnitzii* have also been associated with an increased risk of post-operative recurrence of Crohn disease[47]. *F. prausnitzii* exerts anti-inflammatory effects both in vitro and in vivo in specific pathogen-free and gnotobiotic mice subjected to experimental acute colitis[34,48–50]. The health-beneficial effects of *F. prausnitzii* stem from various mechanisms affecting the epithelial barrier, the local and systemic metabolism, and the immune system. *F. prausnitzii* can stimulate goblet cells to produce mucus glycans, dampens the activation of the NF-κB pathway through the secretion of a 15 kDa soluble microbial anti-inflammatory molecule[51], as well as several metabolites (including short-chain fatty acids[52,53], 4-Hydroxy-butyrate[54], the anti-inflammatory shikimic and salicylic acids, and the osmoprotective raffinose[54]). Importantly, the human colonic mucosa contains *F. prausnitzii*-specific regulatory type 1-like IL-10-secreting and Foxp3-negative T cells that are characterized by a double expression of CD4 and CD8α (DP8α) endowed with potent anti-inflammatory functions[55]. The adoptive transfer of a HLA-DR*0401-restricted DP8α Treg clone combined with *F. prausnitzii* oral supplementation conferred a protective effect against acute colitis in HLA-DR*0401 transgenic NSG model[55]. *F. prausnitzii* directly acted on antigen presenting cells (APC), inducing DC to express anti-inflammatory mediators (such as IL-10, IL-27, CD39, IDO-1, and PDL-1) reducing overt TLR4 signaling[35]. We surmise that these homeostasis-promoting effects of *F. prausnitzii* are seminal to prevent or compensate for the cancer-associated stress ileopathy observed in patients diagnosed with solid tumors[56]. Indeed, the β2 adrenergic receptor-dependent ileal atrophy described in tumor-bearing mice and patients culminates in intestinal dysbiosis (dominated by the *Enterocloster* genus) that contributes to tumor incidence and severity[56].

Besides maintaining homeostasis, *F. prausnitzii* was correlated with the immunostimulatory effects of ICI, increasing blood ICOS⁺CD4⁺ T cells and sCD25 serum levels as well as reprogramming of the tumor microenvironment[28,30]. Of note, *F. prausnitzii* could increase α-ketoglutarate release[54] that in turn, was shown to induce PDL-1 and to synergize with anti-PD-1 antibody[57]. *F. prausnitzii* might also exert direct cytostatic activity on tumor cells in vitro. Its supernatant could inhibit the autocrine secretion of IL-6 and the phosphorylation of JAK2/STAT3 in the breast cancer cell line MCF-7[58]. This effect was ascribed to *F. prausnitzii* metabolites. In a study comparing stool MG and serum

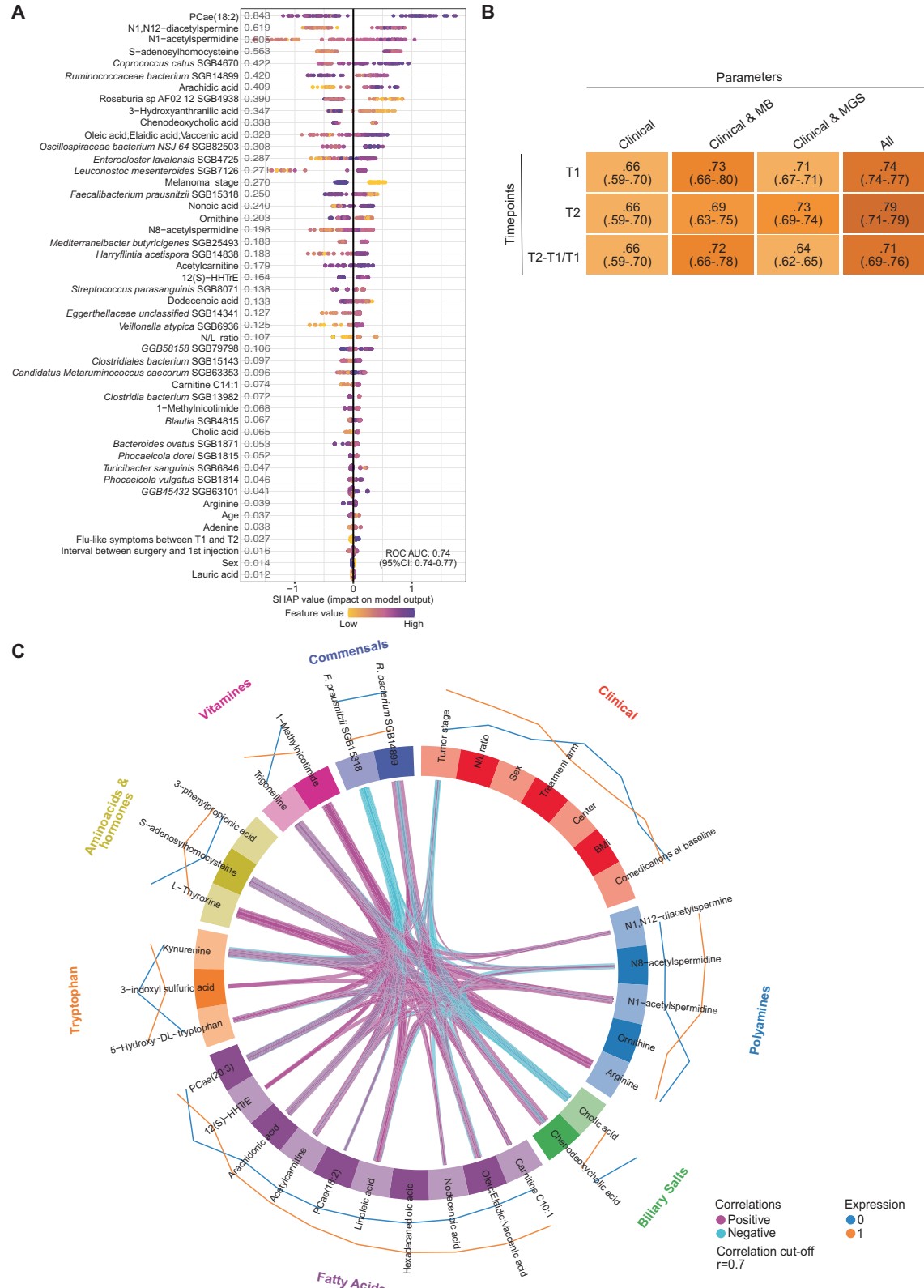

**Fig. 5 | Machine learning (ML, XGBoost) algorithm to identify biomarkers and their interaction predicting recurrence in patients with stage III melanoma.** **A** ML model summary. Features are clinical parameters and metabolomics (MB) + metagenomics (MG) monitored in serum and feces, respectively, at T1 ($n = 88$ patients). SHapley Additive exPlanations (SHAP) values for each feature per patient are positive when the value of the feature increases the prediction of recurrence, negative otherwise. Each dot represents one patient and the color represents the value of each feature. The importance of the feature is depicted with the number on the left column. **B** ML performance using Boruta feature selection algorithm based on XGboost for 2Y-R prediction. Representation of the Area Under the ROC Curve (AUC) values for each treatment arm and feature (clinical, MB or MGS parameters) according to T1, T2 and T2-T1 slope of the trajectory. ROC: receiver operating characteristic. **C** Circosplot indicating correlations between common features described in (**A**, **B**), thickness of lines indicating an increasing positive (pink) or negative (blue) correlation.

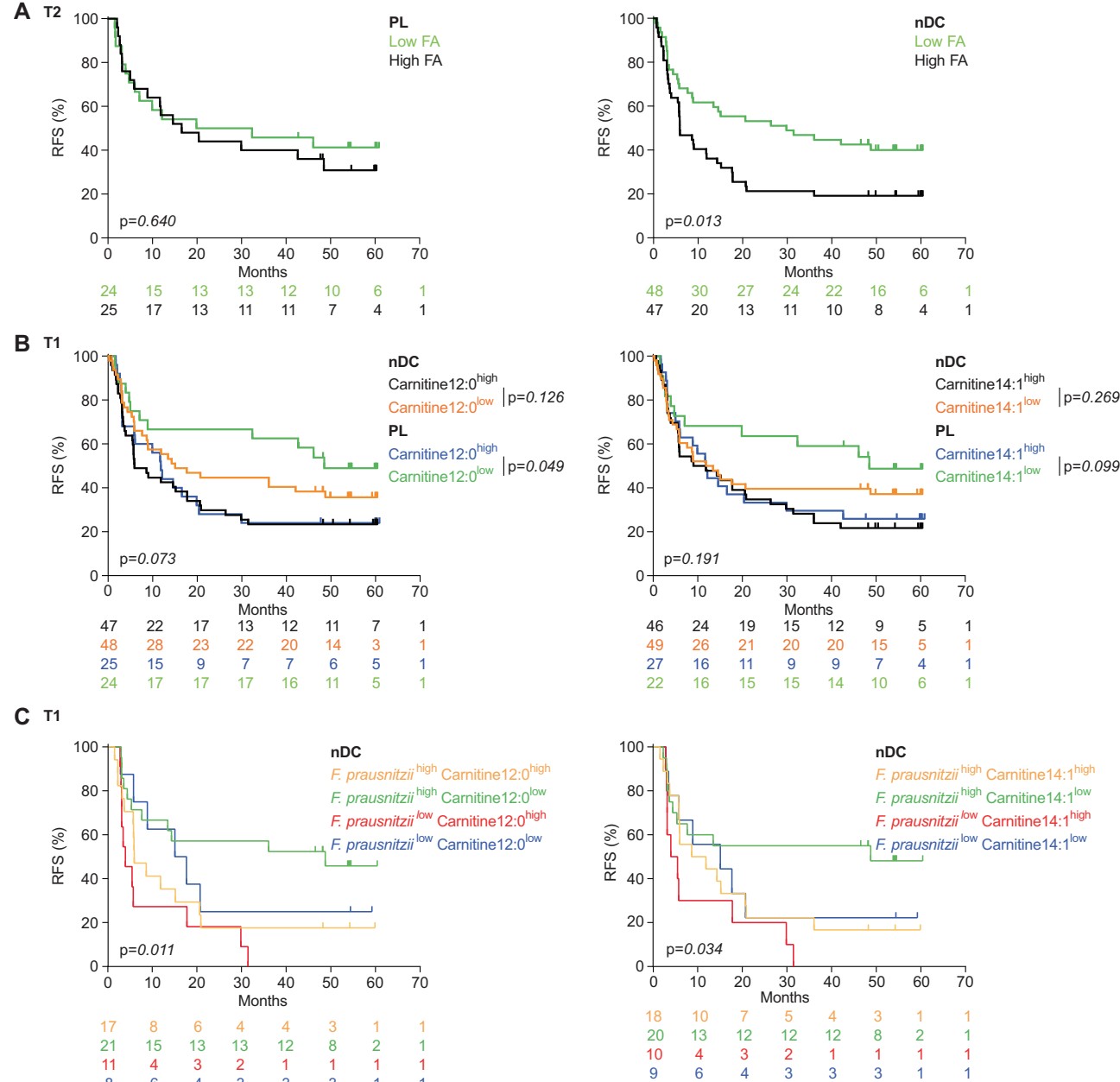

**Fig. 6 | Patient recurrence-free survival (RFS) according to *Faecalibacterium prausnitzii* and fatty acids. A** RFS analysis using the Kaplan–Meier (KM) estimator (Log-Rank (Mantel Cox) test) to assess the predictive value of low abundance of fatty acids (low FA) versus high abundance of fatty acids (high FA) after 2 biweekly injections (T2) in placebo (PL, left panel) and natural Dendritic Cell (nDC, right panel) arms. **B** Same as in (**A**) for Carnitine C12:0 (left panel) and Carnitine C14:1 (right panel) at T1. **C** Same as in (**B**) for Carnitine C12:0 (left panel) and Carnitine C14:1 (right panel) associated with relative abundances of *Faecalibacterium prausnitzii* at T1. The numbers per group are depicted under the KM plots.

MB in 50 patients diagnosed with breast malignant or benign nodules, the authors showed a significant reduction in the relative abundance of *F. prausnitzii* in the breast cancer group. The MB profiles of patients with breast cancer differed from that of controls in the linoleic acid metabolism, and biosynthesis of unsaturated FA[58]. *F. prausnitzii* was also negatively correlated with the levels of phospholipid metabolites, such as 1-stearoyl-2-hydroxy-sn-glycero-3-phosphocholine, 1-Oleoyl-sn-glycero-3-phosphocholine and sphingomyelin.

In fact, accumulating evidence points to a role of *F. prausnitzii* in regulating lipid metabolism. FA are involved in energy metabolism and cell membrane structural components involved in wound healing and cancer cell metabolism[59]. In tumor tissues, free FA are esterified to fatty acyl-CoAs and then transported into the mitochondria by the carnitine shuttling, while in normal tissue, they undergo beta-oxidation as fatty acyl-CoAs to feed into the TCA cycle[60,61]. Patients with Nonalcoholic fatty liver disease (NAFLD) exhibit a decreased relative abundance or prevalence of *F. prausnitzii* compared with HV[62]. Approximately 20–80% of NAFLD patients have dyslipidemia[63]. Treatment with distinct *F. prausnitzii* strains reversed dyslipidemia symptoms in NAFLD mice[53]. Oral gavage with *F. prausnitzii* in high fat diet-treated mice appeared to increase FA oxidation and adiponectin signaling in liver and increased adiponectin expression in visceral adipose tissue[64]. Gas chromatography analysis of hepatic lipid classes revealed a drop in several FA (such as stearic acid, arachidonic acid, eicosapentaenoic

acid, and docosahexanoic acid) in *F. prausnitzii*-treated mice[64]. Our findings highlight the potential critical impact of lipid metabolism in melanoma aggressiveness[65,66]. The contribution of adipose tissue and lipid metabolism, in particular glycerophospholipids, sphingolipids, sterols and eicosanoids in melanoma plasticity has been reviewed[67]. There is a de novo synthesis (of cholesterol, glycerophospholipids, sphingolipids, droplets of neutral lipids), an elongation and desaturation of FA cells as well as importation of FA from neighboring adipocytes that can fuel FA β-oxidation in mitochondria of melanoma cells. De novo lipogenesis together with hypoxia and driver mutations cooperate in melanoma cells to meet their energetic demands. In addition to cell intrinsic pathways, melanoma cells secrete the immunosuppressive PGE2 and the S1P lipids to escape cancer immunosurveillance. Lipid metabolites have a major impact on APC and macrophages. Macrophages are a rich source of bioactive lipids, including FA and their oxygenated forms, oxylipins, following incorporation of lipoproteins or non-esterified FA bound to albumin as well as phagocytosis and efferocytosis. Saturated FA and oxylipins strongly contribute to type 1 inflammation by macrophages (associated with their tumoricidal and proinflammatory phenotype) while IL-4 -mediated M2 polarization tends to dampen FA and oxylipin synthesis[68]. Inhibition of FA synthesis by DC, albeit reducing their differentiation, enhanced their T cell stimulatory capacities, inducing an endoplasmic reticulum stress response associated with MHC class II, costimulatory molecule and cytokine expression[69]. Further, blockade of FA synthesis increased the DC/NK cell cross-talk leading to IFNγ secretion[69]. Two other studies demonstrated that saturated FA activated TLR4, whereas n-3 and n-6 polyunsaturated FA inhibited LPS activation, resulting in altered DC surface molecule expression of CD40 and reduced TNF-α or IL-12p40 release[70,71]. Finally, the liver could also be a source of FA and account for BA deregulation. In addition to absorbing circulating FA, hepatocytes synthesize FA from dietary carbohydrates that reach the hepatocytes via the portal vein[72]. The enterohepatic recirculation of BA exerts important regulatory effects on many hepatic, biliary, and intestinal functions. Dysbiosis can modulate biliary salt composition, leading to a downregulation of MAdCAM-1 and the exodus of immunosuppressive enterotropic T cells promoting tumor growth[73].

Hence, our findings allow to postulate that patients with stage III melanoma of dismal prognosis harbored major deviations in lipid metabolism at diagnosis, likely originating from a diseased enterohepatic axis, that corrupted the immunostimulatory potential and/or exacerbated the tolerogenic functions of autologous nDC in certain patients. These data support the use of distinct biomarkers belonging to specific pathways (listed in the circosplot Fig. 5C) for patient stratification. Prospective trials assessing immunotherapy strategies in melanoma patients should validate the clinical relevance of these biomarkers.

## Methods
The MIND-DC trial (NCT02993315) complies with all relevant ethical regulations (Dutch Central Committee on Research Involving Human Subjects).

### Ethics approval and consent to participate
The study design and conduct complied with all relevant regulations regarding the use of human study participants and was conducted in accordance with the criteria set by the Declaration of Helsinki. Written informed consent was obtained from all patients. Consent to publish clinical information potentially identifying individuals was obtained (age and gender).

### Medical centers
The MIND-DC trial (NCT02993315) was performed in 2 centers in the Netherlands (Radboud University Medical Center, Nijmegen and Isala, Zwolle).

### Trial design
Double-blind, randomized, placebo-controlled phase III clinical trial. Patients with resected stage IIIB and IIIC cutaneous melanoma were randomized in a 2:1 ratio to nDC or PL. Patients received either intra-nodal injections of nDC ($3-8 \times 10^6$/injection) or PL every 2 weeks (biweekly) for 3 doses (one cycle), repeated after 6 and 12 months. The primary endpoint was the 2Y-RFS. One patient was excluded due to insufficient follow-up to avoid censoring bias using binary classifiers. Treatment was stopped in case of disease recurrence (including both loco-regional and distant metastases), unacceptable toxicity, or withdrawn from the study. Details are described in a separate manuscript[27]. Of note, dietary habits and life style -related pieces of information were not collected at the time of protocol conception, restraining some important correlations with MG and MB results.

### Dendritic cell isolation and vaccine preparation
Patients in the nDC arm were vaccinated with autologous nDC loaded with tumor peptides and overlapping peptide pools. Cells were harvested by apheresis and conventional and serumcytoid DC were isolated with the fully automated and enclosed immunomagnetic CliniMACS Prodigy® isolation system (Miltenyi Biotec). nDC were pulsed with MACS® GMP-grade PepTivators®, overlapping peptide pools of the CTA MAGE-A3 and NY-ESO-1 (Miltenyi Biotec) covering the sequence of the entire antigen, and a mix of fourteen peptides of TAA gp100 and tyrosinase and CTA MAGE-C2, MAGE-A3 and NY-ESO-1 (all Leiden University Medical Center, Leiden, the Netherlands) and matured with protamine/mRNA (Meda Pharma, Amstelveen, the Netherlands and Universitätsklinikum Erlangen, Erlangen, Germany).

### Collection of stool and blood samples
Stool samples were prospectively collected at different time points: at the day of the first (T1) and the third (T2) injection (first cycle) at each center following the International Human Microbiome Standards (IHMS) guidelines. Both T1 and T2 samples were considered for this analysis. Blood samples were collected at the same timepoints.

### Metagenomics analysis of patient stools
Overall, 185 fecal samples from 93 patients were sequenced with whole genome sequencing technology. Aliquots of stool samples were stored with DNA/RNA Shield Buffer (Zymo) at −20 °C until use. DNA was extracted from aliquots of fecal samples using the DNeasy PowerSoil Pro Kit (Qiagen) following the manufacturer's instructions. Sequencing libraries were prepared using the Illumina® DNA Prep, (M) Tagmentation kit (Illumina), following the manufacturer's guidelines. A cleaning step on the pool with 0.7x Agencourt AMPure XP beads was implemented. Sequencing was performed on a NovaSeq 6000 S4 flow cell (Illumina) at the internal sequencing facility at University of Trento, Trento, Italy. Raw sequenced reads were QCed using the pipeline available at https://github.com/SegataLab/preprocessing. Briefly, low-quality reads (Q < 20), short reads (< 75 bp), and reads with at least 2 ambiguous bases were discarded. Then, host DNA contaminants were removed (hg19 and phiX174 Illumina spike-in). We then obtained an average of 48 million reads per sample. Twelve samples did not pass internal control and were excluded from the analysis. For each metagenome we profiled the taxonomic and functional potential compositions with MetaPhlAn 4 (database vJan21)[33] and HUMAnN 3.6[74], respectively. For alpha and beta-diversity, we computed the per-sample Shannon index[75] and the between-samples Bray-Curtis dissimilarities using the implementation available in the Vegan R package[76]. Differences in the distributions of alpha and beta diversity for samples collected at diagnosis with respect to recurrence at 2 years were then evaluated using Wilcoxon rank sum test[77]. To test for differential abundance according to recurrence, recurrence at 2 years and treatment, we fitted a generalized linear model for each microbiome

feature via the MaAsLin2 R package[78]. Microbial features are first AST or CLR transformed. Adjusted P-values (Q) are computed via the Benjamini-Hochberg procedure to control for False Discovery Rate (FDR). Prevalence threshold in the differential abundance analysis was set in order to guarantee a minimum number of positive samples in each comparison. In particular, when testing for 2yR and treatment at baseline, we considered a prevalence threshold of 10%, while 30% was used when testing for 2yR, considering each treatment arm independently. We tested for differential abundance SGBs between T1 and T2 via a Wilcoxon signed-rank test considering each treatment arm and 2y-noR/2y-R combination independently. Only SGBs that were present in at least 10 samples in one of the time points are considered in this analysis.

## Metabolomics analysis

**Serum sample preparation and widely targeted detection by LC-MS.** Fifty (50) μl of collected sera were mixed with 500 μl of ice-cold extraction mixture (methanol/water, 9/1, −20 °C, with labeled internal standard). To facilitate endogenous metabolites extraction, samples were then completely homogenized (vortexed 5 min at 2500 rpm) and then centrifuged (10 min at 15,000 g, 4 °C). Supernatants were collected and several fractions were split to be analyzed by different Liquid chromatography coupled with mass spectrometers (LC/MS)[79]. Polyamines and biliary acids analysis were performed by LC-MS/MS with a 1290 UHPLC (Ultra-High Performance Liquid Chromatography) (Agilent Technologies) coupled to a QQQ 6470 (Agilent Technologies). Regarding polyamines analysis, gas temperature was set to 350 °C with a gas flow of 12 l/min. The capillary voltage was set to 2.5 kV. Ten (10) μl of sample were injected on a Column Kinetex C18 (150 mm × 2.1 mm particle size 2.6 μm) from Phenomenex, protected by a guard column C18 (5 mm × 2.1 mm) and heated at 40 °C by a Pelletier oven. The gradient mobile phase consisted of water with 0.1% of Heptafluorobutyric acid (HFBA, Sigma-Aldrich) (A) and acetonitrile with 0.1% of HFBA (B) freshly made. The flow rate was set to 0.4 ml/min, and gradient as follows: initial condition was 95% phase A and 5% phase B. Molecules were then eluted using a gradient from 5% to 30% phase B over 7 min. The column was washed using 90% mobile phase B for 2.25 minutes and equilibrated using 5% mobile phase B for 4 min. The autosampler was kept at 4 °C. Regarding biliary acids analysis, gas temperature was set to 310 °C with a gas flow of 9 L/min. Capillary voltage was set to 4.5 kV. Ten (10) μl of sample were injected on a Column Poroshell 120 EC-C8 1200bars (P/N 981758-902, 100 mm × 2.1 mm particle size 1.9 μm) from Agilent technologies, protected by a guard column XDB-C18 (5 mm × 2.1 mm particle size 1.8 μm) and heated at 40 °C by a Pelletier oven. Gradient mobile phase consisted of water with 0.2% of formic acid (A) and acetonitrile/isopropanol (1/1; v/v) (B) freshly made. Flow rate was set to 0.5 mL/min, and gradient as follow: initial condition was 70% phase A and 30% phase B, changing to 38% phase B over 2 minutes. Phases proportion was still over 2 minutes, then molecules were eluted using a gradient from 38% to 60% phase B over half a minute. Column was washed using 98% mobile phase B for 2 minutes and equilibrated using 30% mobile phase B for 1.5 min. Autosampler was kept at 4 °C. Pseudo-targeted analysis by UHPLC-HRAM (Ultra-High Performance Liquid Chromatography−High Resolution Accurate Mass) was performed on a U3000 (Dionex)/Orbitrap q-Exactive (Thermo) coupling, previously described[80,81]. All targeted treated data were merged and cleaned with a dedicated R (version 4.0) package (@Github/Kroemerlab/GRMeta).

**Data processing and statistical analysis.** Raw data were preprocessed and analyzed with R using the GRMeta package (@Github/Kroemerlab/GRMeta). This software included statistical analysis using a multivariate method approach, as PCA, Heatmap and data visualization, as volcano plots. Area intensity levels were corrected with a quality control pooled sample-based algorithm and normalized area were then log2-transformed prior to heatmaps, boxplots and volcano

plots visualizations. A total of 152 metabolites were finally analyzed for serum samples at T1 and at T2. The best significant metabolites were presented with boxplots with metabolite levels log scaled. Mann−Whitney U-test with no adjustment were conducted on data gathered by two groups on processed data with R. In cases when data treatments were performed on more than two groups of patients, Kruskal-Wallis test followed by a Dunn's test with no adjustment were used on processed data. Pearson correlation analysis was applied on log2 transformed data from metabolite normalized profiles and relative abundances of *F. prausnitzii* SGB15318 and SGB15322. Relevant metabolites correlating with *F. prausnitzii* SGBs were selected and analyzed by enrichment functional analysis with Metaboanalyst (https://www.metaboanalyst.ca) using the KEGG Database[44] for significant metabolites annotation and visualization.

## Statistical analysis

Data analyses were performed with the Prism 10 (GraphPad, San Diego, CA, USA) and the R software. Prism always reports p-values to four decimal places. The prevalence of MGS was calculated using microbial relative abundances (MetaPhlAn 4) and considered absent if relative abundance equal to 0 and present if relative abundance superior to 0. Chi-square test was used for comparison of unpaired groups, considered significant at $p < 0.05$. For MGS, two groups of patients were defined by reference values of relative abundances (MetaPhlAn 4) from publicly available HV cohort[42]: high if ≥ median and low if <median of the MGS relative abundance from HV. For key metabolites, two groups of patients were defined by its abundance median in the overall MIND-DC cohort: high if ≥ median and low if <median. RFS analysis were performed using KM estimator. As the analysis of compositional data can lead to misleading results due to spurious correlation, we used the CLR transform to project the MGS relative abundances from the simplex to the more usual Euclidean space using the *clr* function of the compositions R package.

**Longitudinal analysis.** The analysis of the metabolite or CLR transformed microbial SGB evolution between T1, $M_{T1}$, and T2, $M_{T2}$, was performed using the following linear regression adjusted for the clinical covariables $\boldsymbol{X}$ (age, sex, BMI, stage, and ECOG-PS), considering $M_{T2}$ as response and $M_{T1}$ as offset: $M_{T2} = M_{T1} + \beta_0 + \beta_{DC} + \boldsymbol{\beta X}$. The intercept $\beta_0$ represents evolution of $M$ ($M_{T2} - M_{T1}$) in the PL arm, and $\beta_{DC}$ represents the impact of the nDC (i.e., the difference of the evolution between the two arms). The Wald test $p$ values of $\beta_0$ and $\beta_{DC}$ were provided and considered as statistically significant when $p < 0.05$. The evolution in the nDC arm was estimated by $\beta_0$ reversing the arm reference (i.e., from the model $M_{T2} = M_{T1} + \beta_0 + \beta_{PL} + \boldsymbol{\beta X}$).

**Machine Learning.** All ML models were developed using R. Feature selection and prediction were based alternatively on two different outcomes: the 2Y-R and treatment arm (nDC versus PL arms). For each type of outcome, three datasets per omic (clinical features only, MB, MG, or MB and MG) were defined from the timepoint that was used: T1, T2, and T2-T1/T1 (pre treatment + evolution until T2 i.e., T2-T1) and a third dataset constructed by joining T2-T1 and T1 values. For each model, clinical features were always included in both feature selection and prediction phases. The ML pipeline is based on a first step of feature selection using Boruta feature selection algorithm based on eXtreme Gradient Boosting (XGboost) algorithm[82] to identify most relevant features among clinical and biological markers (both in MB and/or MG). A second step consisted in re-training the model from the subset of selected features re-including clinical variables in order to control potential confounding bias. A last fit of the model using selected biomarkers was applied on our training data, enabling the model to prepare for future label predictions on new data. For the multi-omics model, we performed a second step of feature selection from the metabolites and MGS selected in each omics in the first step (+clinical variables) to obtain our final model. Missing values of metabolites were imputed using the multiple imputation by chained

equations (MICE) method using the mice R package[83]. We performed 50 imputations with 50 iterations to capture the uncertainty of the imputation procedure. The feature selection procedure described above was repeated using the 50 imputed datasets, and the metabolites selected in more than 75% of these 50 iterations were retained for the final step (assuming that they are robust to the randomness of the imputation). A single last imputation was performed to retrain the model in the final step. The model explainer of the different final model was based on the Shapley Additive exPlanations (SHAP) analysis, which is a visualization tool based on the following construction: SHAP values are weights associated to features for each patient, positive when the value of a marker for this patient tends to increase the prediction as class 1 (2Y-R), negative otherwise (2Y-noR). The more the absolute value of SHAP value increases, the more the feature is likely to impact the prediction (as class 1 if SHAP_value > 0, class 0 otherwise). SHAP values are thus positive or negative continuous values.

Correlations between biomarkers were analyzed via the mixOmics package in R[84], using the DIABLO multiblock sPLS-DA method to display explanatory relationship between pathways and then displayed into a circosplot. The prediction performance of the whole model pipeline (feature selection to model fitting) was evaluated using the bootstrap optimism corrected area under the receiver operating characteristic (AUROC) curve. Due to the lack of external validation cohort, the optimism of the AUROC was corrected using the 0.632+ bootstrap method[85]. The confidence interval of these optimism corrected AUROC was obtained using two-stage bootstrap methods proposed by ref. 86 (50 internal samples, 500 external samples). The 0.632+ estimators are displayed on each figure.

## Reporting summary

Further information on research design is available in the Nature Portfolio Reporting Summary linked to this article.

## Data availability

The Metagenomics data generated in this study have been deposited in the European Nucleotide Archive (ENA) database under accession code PRJEB66197. The Metabolomics data generated in this study have been deposited in the Mendeley Data database under accession code DOI 10.17632/nzb653783h.1 [https://data.mendeley.com/datasets/nzb653783h/1][87]. Further individual participant clinical trial data are available under restricted access for privacy and ethical restrictions, access can be obtained by contacting the corresponding author of the companion paper[27] (Dr. I. Jolanda M. de Vries, e-mail address Jolanda.deVries@radboudumc.nl). Data requests will be reviewed by the principal investigators of the trial. Any data and materials that can be shared will require approval from the Institutional Review Board and a data or material transfer agreement. The remaining data are available within the Article, Supplementary Information or Source Data file. Source data are provided with this paper.

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

## Acknowledgements

We would like to particularly thank the patients who participated in the trial and all employees at the Radboud university medical center and Isala hospital who were involved in the trial execution. The MIND-DC trial was funded by ZonMw, Ministry of Health, Welfare and Sport (VWS), Stichting ATK, Miltenyi Biotec (in-kind). This work was supported by SEERAVE Foundation, European Union Horizon 2020:Project Number: 825410 and Project Acronym: ONCOBIOME, Institut National du Cancer (INCa), ANR Ileobiome – 19-CE15-0029-01, ANR RHU5 "ANR-21-RHUS-0017" IMMUNOLIFE", MAdCAM INCA_ 16698, Ligue contre le cancer, LABEX OncoImmunology, la direction generale de l'offre de soins (DGOS), Universite Paris-Sud, SIRIC SOCRATE (INCa/DGOS/INSERM 6043), and PACRI network. G.K. is supported by the Ligue contre le Cancer (équipe labellisée); Agence National de la Recherche (ANR) – Projets blancs; AMMICa US23/CNRS UMS3655; Association pour la recherche sur le cancer (ARC); Cancéropôle Ile-de-France; Fondation pour la Recherche Médicale (FRM); a donation by Elior; Equipex Onco-Pheno-Screen; European Joint Programme on Rare Diseases (EJPRD); European Research Council Advanced Investigator Award (ERC-2021-ADG, ICD-Cancer, Grant No. 101052444), European Union Horizon 2020 Projects Oncobiome, Prevalung (grant No. 101095604) and Crimson; Fondation Carrefour; Institut National du Cancer (INCa); Institut Universitaire de France; LabEx Immuno-Oncology (ANR-18-IDEX-0001); a Cancer Research ASPIRE Award from the Mark Foundation; the RHU Immunolife; Seerave Foundation; SIRIC Stratified Oncology Cell DNA Repair and Tumor Immune Elimination (SOCRATE); and SIRIC Cancer Research and Personalized Medicine (CARPEM). This study contributes to the IdEx Université de Paris ANR-18-IDEX-0001. This work is supported by the Prism project funded by the Agence Nationale de la Recherche under grant number ANR-18-IBHU-0002. CACS was funded by MSD Avenir. MF is funded by SEERAVE Foundation and MERCK Foundation. LD and BR were supported by Philantropia at Gustave Roussy Foundation. The funders had no role in the design of the study, in the writing of the manuscript, or in the decision to publish the results.

## Author contributions

Conceptualization: C.A.C.S., G.K., L.Zi., L.D. K.F.B. and I.J.M.V. Methodology: D.D., D.S, G.S., C.A.C.S., M.B., G.P., S.D., F.Ap., M.P., F.As., F.P., N.S., G.K. and L.Zi. Software: G.P., D.S., M.B., S.D., P.M. and D.D. Validation: C.A.C.S., G.P., D.S., M.B., G.S., S.D., P.M. and D.D. Formal analysis: C.A.C.S., G.P., D.S., M.B., S.D., D.D. and L.D. Investigation: G.S., C.A.C.S., M.B., G.P., K.F.B., S.D., M.F., R.B. and S.T. Resources: K.F.B., I. J.M.V., G.S., G.K., L.Zi. and L.D. Data Curation: G.P., M.B., G.S., S.D., F.Ap., P.M., M.P., F.As., F.P., F.Ar., K.F.B. and I.J.M.V. Writing - Original Draft: L.Zi. and C.A.C.S. Writing - Review & Editing: all but in particular C.A.C.S., G.P., M.B., K.F.B., S.D., M.F., C.S., B.R., A.M.M.E., G.K., I.J.M.V., L.Zi. and L.D. Visualization: C.A.C.S. Supervision: N.S., L.Zi., L.D., K.F.B. and I.J.M.V. Project administration: C.A.C.S., C.S., N.S., G.K., L.Zi., L.D., K.F.B. and I.J.M.V. Funding acquisition: G.K., I.J.M.V., L.Zi. and L.D.

## Competing interests

L.Zi. founded EverImmune and is the SAB president of EverImmune. L.ZI. had grant support from Daichi Sankyo, Kaleido, 9 meters and Pileje. A.M.M.E.: consulting fees from Acetra, Agenus, BioInvent, Brenus, CatalYm, Epics, Ellipses, Galecto, GenOway, IO Biotech, IQVIA, ISA Pharmaceuticals, Merck&Co, MSD, Pierre Fabre, Sairopa, Scorpion, Sellas, SkylineDX, TigeTx, Trained Immunity TX. A.M.M.E.: participation on a Data Safety Monitoring Board: Boehringer Ingelheim, BioNTech, and Pfizer. A.M.M.E.: lectures for BMS, MSD. A.M.M.E.: stock or stock options for IO Biotech, SkylineDx and Sairopa. G.K. has been holding research contracts with Daiichi Sankyo, Eleor, Kaleido, Lytix Pharma, PharmaMar, Osasuna Therapeutics, Samsara Therapeutics, Sanofi, Tollys, and Vascage. G.K. is on the Board of Directors of the Bristol Myers Squibb Foundation France. G.K. is a scientific co-founder of EverImmune, Osasuna Therapeutics, Samsara Therapeutics and Therafast Bio. G.K. is in the scientific advisory boards of Hevolution, Institut Servier and Longevity Vision Funds. G.K. is the inventor of patents covering therapeutic targeting of aging, cancer, cystic fibrosis and metabolic disorders. G.K.'s brother, Romano Kroemer, was an employee of Sanofi and now consults for Boehringer-Ingelheim. B.R. is co-founder of Science Curebiota. L.D. is a SAB member of EverImmune. K.F.B.: consulting fees (institutional) from MSD and Pierre Fabre. The remaining authors declare no competing interests.

## Additional information

Carolina Alves Costa Silva [1,2,3,19], Gianmarco Piccinno [4,19], Déborah Suissa[1,2,3,19], Mélanie Bourgin [5,6,19], Gerty Schreibelt [7,19], Sylvère Durand[5,6], Roxanne Birebent[1,2], Marine Fidelle [1,3], Cissé Sow[1,3], Fanny Aprahamian[5,6], Paolo Manghi [4], Michal Punčochář [4], Francesco Asnicar [4], Federica Pinto[4], Federica Armanini[4], Safae Terrisse[8], Bertrand Routy [9,10], Damien Drubay [1,11,12], Alexander M. M. Eggermont[13,14], Guido Kroemer[5,6,15], Nicola Segata [4,16], Laurence Zitvogel [1,2,3,17,20] ✉, Lisa Derosa[1,2,3,20], Kalijn F. Bol[7,18,20] & I. Jolanda M. de Vries [7,20]

[1]Gustave Roussy Cancer Campus (GRCC), ClinicObiome, Villejuif Cedex, France. [2]Faculté de Médecine, Université Paris-Saclay, Kremlin-Bicêtre, France. [3]Institut National de la Santé Et de la Recherche Médicale (INSERM) U1015, Équipe Labellisée – Ligue Nationale contre le Cancer, Villejuif, France. [4]Department of Computational, Cellular and Integrative Biology (CIBIO), University of Trento, Trento, Italy. [5]Metabolomics and Cell Biology Platforms, Gustave Roussy Cancer Campus, Villejuif, France. [6]Centre de Recherche des Cordeliers, INSERM U1138, Équipe Labellisée – Ligue Nationale contre le Cancer, Université Paris Cité, Sorbonne Université, Paris, France. [7]Medical BioSciences, Radboud Institute for Medical Innovation, Radboud university medical center, Nijmegen, The Netherlands. [8]Oncology Department, Assistance Publique Hôpitaux de Paris (AP-HP), Hôpital Saint-Louis, Paris, France. [9]Centre de Recherche du Centre Hospitalier de l'Université de Montréal (CRCHUM), Montréal, QC, Canada. [10]Hematology-Oncology Division, Department of Medicine, Centre Hospitalier de l'Université de Montréal (CHUM), Montréal, QC, Canada. [11]Office of Biostatistics and Epidemiology, Gustave Roussy Cancer Campus, Université Paris-Saclay, Villejuif, France. [12]Inserm, Université Paris-Saclay, CESP U1018, Oncostat, labeled Ligue Contre le Cancer, Villejuif, France. [13]Princess Máxima Center and University Medical Center Utrecht, 3584 CS Utrecht, The Netherlands. [14]Comprehensive Cancer Center Munich, Technical University Munich & Ludwig Maximiliaan University, Munich, Germany. [15]Department of Biology, Institut du Cancer Paris CARPEM, Hôpital Européen Georges Pompidou, AP-HP, Paris, France. [16]Department of Experimental Oncology, IEO European Institute of Oncology IRCCS, Milan, Italy. [17]Center of Clinical Investigations BIOTHERIS, INSERM CIC1428 Villejuif, France. [18]Department of Medical Oncology, Radboud university medical center, Nijmegen, The Netherlands. [19]These authors contributed equally: Carolina Alves Costa Silva, Gianmarco Piccinno, Déborah Suissa, Mélanie Bourgin, Gerty Schreibelt. [20]These authors jointly supervised this work: Laurence Zitvogel, Lisa Derosa, Kalijn F. Bol, I. Jolanda M. de Vries. ✉e-mail: laurence.zitvogel@gustaveroussy.fr

