## [Peer Review File · Nature Communications]

Influence of microbiota-associated metabolic reprogramming on clinical outcome in patients with melanoma from the randomized adjuvant dendritic cell-based MIND-DC trialEditorial Note: Parts of this Peer Review File have been redacted as indicated to remove third-party material where no permission to publish could be obtained.

REVIEWER COMMENTS

Reviewer #1 (Remarks to the Author): with expertise in microbiome, cancer

In the present manuscript, the authors describe associations of the gut microbiome and serum metabolome with clinical outcomes in the MIND-DC phase III clinical trial designed as adjuvant trial in stage III melanoma patients. The trial design comprises a 2:1 randomization of intranodal injections of pulsed dendritic cells vs. placebo. In the current paper of a back-to-back submission with a clinical study on the efficacy of DC immunotherapy, the authors study gut microbiome and serum metabolome profiles associated with 2-years disease-free survival (DFS) rates. They observed that *Faecalibacterium prausnitzii* and serum acylcarnitines to be associated with DFS. However, it turned out that there were a priori differences in stool metagenomes and serum metabolites between the two treatments despite subject randomization.

Overall, the manuscript is well written, all methods are state-of-art and analyses were thoroughly carried out. It is of great value to see that the authors looked into baseline differences between the two intervention groups and observed these remarkable group differences for microbiomes and metabolites. Despite these differences, they were able to detect biomarkers associated with treatment outcomes. Although the overall clinical value of DC-based cellular therapies in adjuvant melanoma therapies needs to be further debated (esp. as it performed slightly worse compared to placebo injections), these current data are important for the cancer immunotherapy field as they underscore the importance of microbiome-host interactions in cancer therapy.

Major points to consider:

The data of the present study can only be interpreted in the context of the intervention study. Therefore, it is of critical importance to ensure that both manuscripts get published at the same time, or, alternatively, get merged into one manuscript.

It would be of great help to understand why these two intervention groups differ in baseline microbiome profiles, and I would recommend additional analyses that include medication (metformin, antibiotic exposures, psychotropic drugs etc.), diet, BMI, previous lines of therapy etc. Are there center effects that could have an impact here, or are there differences in the time periods between surgery and adjuvant DC / placebo injections? Along these lines, what could drive the differences in microbiome and metabolome profiles between T1 and T2 (e.g., antibiotic treatments [see flu like symptoms in the adverse event profiles] etc.).

Regarding the intervention group differences, I would suggest to also analyze and present baseline and T2 taxa and metabolite differences for the 2YR-DFS-outcomes separately (e.g., LM coefficients per intervention). It may be that FP is associated to outcome only in the DC group, although the primary analyses were done with correction by treatment arm in Maaslin. It may also be that the dysbalance between intervention group Ns accounts for the differences? I would also suggest to add rel. abundance plots for the core taxa set to get an impression of the abundances between intervention and outcome groups.

It would be interesting to see AUROCs for 2-YR DFS prediction based on taxa +/- metabolites +/- clinical covariates for PL vs DC.

Minor points:

- Color coding of the PCAs as the color codes for 2YR-R and -NR are a bit misleading.
- Figure 2S: Any recurrence during the follow-up; please specify.
- Figure 1: please add the Ns per group that were finally analyzed for the 2YR-follow-ups.
- Are there differences in overall survival rates as well per fecal taxa and/or serum metabolites?

Reviewer #2 (Remarks to the Author): with expertise in microbiome, cancer

The report by Alves Costa Silva, C. et al aims to describe associations between fecal microbiome and blood metabolomic signatures with clinical outcomes in a randomized DC based clinical trial in patients with stage III melanoma. This report adds to the growing body of literature suggesting links between the composition of the gut microbiome and immune activation that aim to explain heterogenous responses to different types of cancer therapies including immunotherapy. The trial design, sample collection, and data generation presented herein offer numerous opportunities to evaluate associations between gut microbiomes and clinical outcomes; however, the results presented in this report are vastly limited by the analytical approach taken by the authors.

The main limitation of this work is the lack of evaluation of the data in a longitudinal manner. T1 and T2 timepoints are presented in cross-sectional analyses but inaccurately labeled as longitudinal. The longitudinal sample collection of and evaluation of the stool and serum samples are a major highlight of this trial design, and the data should be evaluated in the same fashion. The authors reported differences in baselines between arms but failed to evaluate the data longitudinally within each arm to avoid the biases reported. Evaluating similarities and differences between groups at each timepoint is important, but given this is an interventional trial, evaluating pre vs. post within group and by response status in a paired manner and across groups will yield more meaningful results. This way, the authors could either use the baseline samples to evaluate predictor potential of response of baseline microbiome metrics as well as stability of microbiome over time comparing active arm vs. placebo. Moreover, the authors could derive stability measurements between T1 and T2 and determine if stability is associated with outcomes of interest.

All the p-values presented need to be corrected for multiple comparisons and statistical tests (instead of "model coefficient" report the test used) need to be mentioned alongside p values throughout the text and in figure legends.

Metagenomic and metabolomic and other data reported in this manuscript should be deposited prior to final submission and accession numbers provided in the manuscript. Additionally, data can be made available through supplementary tables.

A full revision of all figures to check for y- and x-axes labels is required as well as figure legends. Labels should not be the sample size for each arm but rather descriptive titles for each group compared.

I highly recommend having a statistician review and revise the manuscript and figures.

Minor Comments

- Check for grammar, syntax,
- Words like “and”, “bacterium”, “spp.”, “model coefficient”, and numbers, including p values, –should not be italicized.
- Avoid words and phrases like “some”, “similar to that”,
- Instead, or in addition of using “significantly correlated”, “a direct correlation”, report a p-value, test, and citation.
- In the abstract, I recommend rewording the statement below in the abstract to convey an accurate perspective of where the field is regards to using (fecal) microbiome composition as a predictor of response as a predictor for survival in melanoma. At best, we have observed associations between fecal microbiome composition and survival metrics which carry more power at a cohort-level than when multiple cohorts are combined.

“The taxonomic composition of the gut microbiome earned its credentials among predictors of survival in melanoma, by influencing the peripheral and tumoral immune tonus.”

- The sentences referring to Sipuleucel-T in prostate cancer in the introduction is not supporting any of the information presented thereafter. I suggest removing or addressing in the discussion in the context of the results presented.
- Here, we investigated.... could modulate “fecal” metagenomic and metabolomic profiles...
- These sentences are not results and should be placed in the discussion:

“In a meta-analysis incorporating new cohorts....1-year progression free survival rates. Bacteria associated ... and Blautia spp.”

“FA are involved in energy metabolism...and cancer cell metabolism. In tumor tissues, free FA...to feed into the TCA cycle”.

- Avoid using terms such as “for the first time”. This is not the first-time shotgun metagenomic sequencing has been used to profiles stool samples from patients with late-stage melanoma.
- This is incorrect: beta diversity of fecal taxa distribution – please reword.
- Citations missing for KEGG, MetaCyc, Bray-Curtis, MetaPhlan4, HUMAnN3.6, Shannon diversity index, VEGAN, Wilcoxon test, Spearman test,
- For the following statement:
“Here, we infer from these findings... with melanoma recurrence”. Instead, present this inference as a hypothesis or a model in the discussion.
- In the section “Taxonomic and metabolomic differences between treatment arms at randomization” – please reword the opening statement. Should not begin with “Hence” since it is the opening of a new results section.
- Replace “monitored” with “assess with”, “evaluated by”, “determined by” in “(monitored by Bray-Curtis dissimilarity)”
- The Spearman test is used to determine if correlations exist, but it does not offer multi-omic data integration. Please check and reword statements.
- Box plots should show all data points to visualize the distribution of data like how it is presented in Figure 2D.

Reviewer #3 (Remarks to the Author): with expertise in melanoma, immunotherapy, cancer

This translational science paper describes why the authors believe the microbiota is responsible for the negative results in a recent phase III clinical trial.

1. In general, the language of the text could be significantly shortened. For example, the first two pages of the results section could be made into a Table making it easier for the reader to follow.
2. What type of supervised hierarchical clustering was utilized for the metabolomics experiments?

3. This is an association study. What mechanism can the authors deduce from their studies?

There will not be definitive experiments, but the patterns of MB and MG should be associated with T and DC function/biochemical processes (or even perhaps B cell function) associated with good responses in melanoma. This reviewer sees hints of this analysis, but it should be further developed.

4. Health-related microbiota other than. Recommend to make a table of the health related microbiota in addition to *F. prausnitzii*. If one did sub stratify the trial population in an exploratory analysis, does removing this confounder variable make a difference?

5. I Would recommend providing T and N staging given AJCC version 7.

6. A pathway map would be expected synthesizing all the data to illustrate the biochemical principles and their relationship to DC and T cell biology. This should be a readable pathway map describing the major pathways and why the authors' hypotheses about microbiota may be correct. This reviewer understands there will not be definitive proof but more significant associations are needed.

Villejuif-Grand-Paris,
September 16th, 2023

REVIEWER COMMENTS

Reviewer #1 (Remarks to the Author): with expertise in microbiome, cancer

In the present manuscript, the authors describe associations of the gut microbiome and serum metabolome with clinical outcomes in the MIND-DC phase III clinical trial designed as adjuvant trial in stage III melanoma patients. The trial design comprises a 2:1 randomization of intranodal injections of pulsed dendritic cells vs. placebo. In the current paper of a back-to-back submission with a clinical study on the efficacy of DC immunotherapy, the authors study gut microbiome and serum metabolome profiles associated with 2-years disease-free survival (DFS) rates. They observed that *Faecalibacterium prausnitzii* and serum acylcarnitines to be associated with DFS. However, it turned out that there were a priori differences in stool metagenomes and serum metabolites between the two treatments despite subject randomization.

Overall, the manuscript is well written, all methods are state-of-art and analyses were thoroughly carried out. It is of great value to see that the authors looked into baseline differences between the two intervention groups and observed these remarkable group differences for microbiomes and metabolites. Despite these differences, they were able to detect biomarkers associated with treatment outcomes. Although the overall clinical value of DC-based cellular therapies in adjuvant melanoma therapies needs to be further debated (esp. as it performed slightly worse compared to placebo injections), these current data are important for the cancer immunotherapy field as they underscore the importance of microbiome-host interactions in cancer therapy.

Major points to consider:

The data of the present study can only be interpreted in the context of the intervention study. Therefore, it is of critical importance to ensure that both manuscripts get published at the same time, or, alternatively, get merged into one manuscript.

Our response:

We agree with this view. The two papers have been resubmitted together.

Q1: It would be of great help to understand why these two intervention groups differ in baseline microbiome profiles, and I would recommend additional analyses that include medication (metformin, antibiotic exposures, psychotropic drugs etc.), diet, BMI, previous lines of therapy etc. Are there center effects that could have an impact here, or are there differences in the time periods between surgery and adjuvant DC / placebo injections?

Our response:

MIND-DC is a double-blinded and placebo-controlled trial. Patients were centrally randomized and stratified according to stage of the disease and adjuvant radiotherapy. A minimization technique was used for random treatment allocation stratifying by stage of the disease (IIIB versus IIIC), adjuvant radiotherapy (planned/received versus not planned/not received), BRAF mutation status (BRAF wildtype versus BRAF mutation (versus unknown)). Clinical characteristics were well balanced as shown in our updated and revised Table S1 to address these questions. There are no differences regarding comedications at baseline, BMI, center and time periods between surgery and adjuvant DC or placebo. Patients have not received any previous systemic therapy before the trial intervention. Sadly, dietary habits and life style-related pieces of information were not collected at that time (back 8 years ago when conceiving the study). So, we conclude there was no overt clinical bias in the randomization process.

Q2: Along these lines, what could drive the differences in microbiome and metabolome profiles between T1 and T2 (e.g., antibiotic treatments [see flu like symptoms in the adverse event profiles] etc.).

Our response:

Overall, only 17/144 (12%) patients started new comedications between T1 and T2. From these, 12/95 (13%) patients were in the DC arm and 5/59 (10%) were in the placebo arm. Only 4 (3%)

patients took ATB between T1 and T2, all in the DC arm. Regarding flu-like symptoms, 28/144 (19%) patients experienced those symptoms between T1 and T2, mostly grade 1 (26/28, 93%) and in the DC arm (23/95=24% versus 10% in the control arm, which is not significantly different). Table S2 depicts the number of patients who started new comedications and experienced flu-like symptoms between T1 and T2. No statistically significant differences were observed (new Table S4 in the revised version).

Q3: Regarding the intervention group differences, I would suggest to also analyze and present baseline and T2 taxa and metabolite differences for the 2YR-DFS-outcomes separately (e.g., LM coefficients per intervention). It may be that FP is associated to outcome only in the DC group, although the primary analyses were done with correction by treatment arm in Maaslin. It may also be that the dysbalance between intervention group Ns accounts for the differences? I would also suggest to add rel. abundance plots for the core taxa set to get an impression of the abundances between intervention and outcome groups.

Our reply:

We thank the reviewer for her/his constructive remarks. By splitting the whole cohort into 4 groups according to treatment arm (nDC arm versus placebo arm “ nDC vs PL”) and response to treatment (2 year-recurrence (2Y-R), 2 year-no recurrence (2Y-noR)) at two time points (baseline (T1) vs 1 month post- therapy (T2)), we obtained smaller sample size (Table S2), with 66% (n=60) and 48% (n=31) of 2Y-R for nDC and PL respectively, and 15% versus 3% of death (9/60 in nDC and 1/31 in PL). This reduces dramatically the statistical power, and therefore the number of significant associations for the metagenomics analyses. Linear model coefficients (MaAsLin2, coefficient) for microbial SGBs that are found associated (after arcsin-sqrt (AST) OR centered-log-ratio (CLR) transformation) with 2Y-R with $P < 0.05$ and $Q > 0.2$ are detailed in Figure S1, Figure S2 and Figure S3 at T1 and T2.

In brief, there is a weak association of *F. prausnitzii* SGB15318 with the prognosis (2Y-noR) in the nDC treatment arm (Figure S1-S2), while this species is not significantly impacted by the nDC therapy (Figure S3, new Figure S4). Of note, *S. salivarius* SGB8007 and *S. parasanguinis* SGB8071 follow the same behavior only in the nDC treatment arm (Figure S1-S2). This result is in line with Figure 3D which shows Kaplan-Meier survival curves indicating that nDC therapy tends to be associated with dismal prognosis only in the subgroup of individuals who have low stool *F. prausnitzii* relative abundances. In addition, the prevalence

and relative abundances of distinct strains of *F. prausnitzii* tend to be reduced in patients from the nDC arm compared with the PL arm (Figure S2C-D) at both time points.

Moreover, few weak ($P < 0.05$, and Q (FDR corrected P) > 0.2) additional changes occurred at microbiome levels overtime during DC therapy. Patients classified 2Y-noR at T2 tended to acquire a relative over-overabundance of species that were associated with no recurrence in the PL arm (such as *B. ovatus*), the best arm in terms of overall survival. In contrast, 2Y-R patients receiving nDC appeared to lose beneficial MGS like *Barnesiella intestinihominis* (that was associated with 2Y-noR in the PL) while acquiring harmful MGS such as *Candidatus Cibiobacter quibialis* (that was lost in 2Y-noR in the PL arm) (Figure S3).

The set of SGBs associated with 2Y-R for the PL arm is instead different (with *Bacteroides ovatus* and *L. pectinoschiza* positively and negatively associated with 2Y-noR respectively). Importantly, when considering events of death at 2 years (not shown, clinicians preferring to stick to 2Y-RFS outcome), we corroborated the association between *F. prausnitzii* strains and no death.

The biomarker evolution between the two timepoints was assessed using linear regression adjusted for the trial arm, age, gender, tumor stage (III-b vs III-c), ECOG performance status, and BMI (see details in the new methods section). We concluded that the nDC treatment arm barely impacted on the evolution of the MG taxonomic composition, except for a few taxa, such as the beneficial commensal *Coprococcus eutactus* and harmful *Enterocloster bolteae* that increased and decreased respectively in nDC patient stools ($p=0.039$ and $p=0.01$ respectively) while remaining stable in the PL arm ($p=0.27$ and $p=0.17$ respectively), independently of age, gender, stage and ECOG-PS (new Figure S4A). Of note, after FDR correction, these evolutions lost their statistical significance. We may impute these weak associations to the small sample size (limiting the power of the analysis) regarding to the data dimensionality (increasing the risk of false positive results, leading to strong penalization of the FDR correction) and to the lack of clear effect of the treatment arm in this negative trial.

We have clearly stated in the text of the revised version that these associations, although meaningful and supported by the literature, remained weak (don't pass the FDR correction) and need prospective validations (discussion section and results section).

Regarding the metabolomics data, the results are now strengthened in the revised version with new findings resulting from the implementation of a different statistical method. We used a machine learning statistical approach taking into account the interactions between these

interdependent factors (on the contrary to the standard multivariable approach that assumes independence of the predictors/explanatory factors). This method named “XGBoost” (eXtreme Gradient Boosting) classification algorithm, was coupled to a well recognized selection strategy (Boruta) to identify the most relevant metabolic/metagenomic or clinical parameter (age, gender, BMI, staging, etc...) to predict the patient response. Because of the black-box nature of this algorithm, we used a model explainer based on SHapley Additive exPlanations (SHAP) values to “open” this black box, and provide a means to the selected clinical and molecular features.

Firstly, we corroborated that biliary acids (BA) (in particular cholic acid and chenodeoxycholic acid at T1&T2 and glycochenodeoxycholic acid at T2) as well as fatty acids and acetylcarnitines (and a few pathways meaningful in other studies including Tryptophan metabolism and vit B3 (trigonelline, 1-methylnicotinamide) and polyamines (acetylated metabolites of spermine/spermidine)) were selected as predictors of the RFS (in accordance to the standard statistical analysis) and interact inbetween each other (novel Figure 5A, AUC=0.72, circosplot in Figure 5C).

Secondly, using the XGBoost/SHAP algorithm model, we could corroborate that the nDC and PL groups differed at baseline for the serum composition of BA with a relative overabundance of primary BA and a lower abundance of ornithine in the nDC patients (revised panel 4B, Figure S7).

Thirdly, the metabolite evolution between the two timepoints (noted T2-T1/T1 in the revised version of the manuscript, and constructed by joining T2-T1 and T1 values) was analyzed by XGBoost, indicating that the increase of FA (oleic acid) and acetylated polyamines (N8 acetylspermidine) and the decrease of ornithine (upstream of the polyamine pathway) and primary BA (glyco-cholic acid, chenodeoxycholic acid) were associated with an increased risk of 2Y-R (Figure S8A). The metabolite evolution between the two timepoints according to the treatment arm was assessed using linear regression adjusted for the trial arm, age, gender, tumor stage (III-b vs III-c), ECOG performance status (details in the methods section), and BMI, revealing very few and weak nDC-associated modulations (Figure S8B).

We applied the same analysis integrating metagenomics and clinical features, obtaining better prediction of 2Y survival at T2 and confirming the prognostic impact of *F. prausnitzii* (Figure 5B). The circosplot highlights the anticorrelations between these MGS and the composition of BA and polyamines (Figure 5C).

The results of the machine learning prediction model integrating all clinical, MG and MB features at T1 are all presented in a new figure, i.e Figure 5A-D (refer to Point-by-point reply Figure 1 below).

We concluded that biliary acids and polyamines are important metabolites associated with the prognosis of stage III melanoma and that significant bias in their relative abundances could discriminate the nDC from the PL group at baseline, while DC did not significantly alter their levels overtime.

Altogether, we added 5 supplemental figures and Figure 5 (predictive model summary and circosplot of cross-correlations) to address these questions in the revised manuscript (refer to PBPR Figure 1).

Figure 5

PBPR Figure 1 (Figure 5 of the revised version). Machine learning (ML, XGBoost) algorithm to identify biomarkers and their interaction predicting survival in stage III melanoma.

A-B. ML model summary. Features are clinical parameters and MB (A) or MG (B) monitored in serum at T1. SHAP values for each feature per patient are positive when the value of the feature increases the prediction of recurrence, negative otherwise. Each dot represents one patient and the color represents the value of each feature. The importance of the feature is depicted with the number on the left column. **C.** Circosplot indicating correlations between common features described in A. and B., thickness of lines indicating an increasing positive (pink) or negative (blue) correlation. **D.** ML model summary. Features are clinical parameters and MB+MG monitored in serum at T1. SHAP values for each feature per patient are positive when the value of the feature increases the prediction of recurrence, negative otherwise. Each dot represents one patient and the color represents the value of each feature. The importance of the feature is depicted with the number on the left column.

Q4: It would be interesting to see AUROCs for 2-YR DFS prediction based on taxa +/- metabolites +/- clinical covariates for PL vs DC.

Our reply:

Using the XGboost algorithm model, we calculated these AUC+CI values (considering bootstrap optimism correction to limit overinterpretation without external validation set due to overfitting) that the reader can now discover in Figure S8C of the revised version (refer to PBPR Figure 2 below).

		Parameters			
		Clinical	Clinical & MB	Clinical & MGS	All
Timepoints	T1	.66 (.59-.70)	.73 (.66-.80)	.67 (.66-.74)	.75 (.68-.80)
	T2	.66 (.59-.70)	.69 (.63-.75)	.70 (.63-.74)	.78 (.68-.79)
	T2-T1/T1	.66 (.59-.70)	.72 (.66-.78)	.67 (.66-.75)	.76 (.66-.78)

PBPR Figure 2 (Figure S8C of the revised version). Interactive statistical machine learning (ML) algorithm to dissect independent biomarkers predicting survival in stage III melanoma. Machine learning predictive values of 2Y-R based on the XGboost/SHAP algorithm model. Representation of the AUC values for each treatment arm and feature (clinical, MB or MGS parameters) according to T1, T2 and T2-T1/T1 slope of the trajectory. ROC AUC for clinical model is identical for T1, T2 and T2-T1/T1 due to a single clinical dataset.

Minor points:

Color coding of the PCAs as the color codes for 2Y-R and -NR are a bit misleading.

Our reply:

Color coding were adjusted to avoid confusion.

- Figure 2S: Any recurrence during the follow-up; please specify. Figure 1: please add the Ns per group that were finally analyzed for the 2YR-follow-ups.

Our reply:

A new Table S3 indicates patient characteristics with the complete follow up at 56 months. In brief, it shows 66% (n=60) and 48% (n=31) of recurrences for DC and PL respectively, and 15% versus 3% of deaths respectively (9/60 in DC and 1/31 in PL). We have updated and changed

the set of Fig S1 to Fig S5 figures and legends.

- Are there differences in overall survival rates as well per fecal taxa and/or serum metabolites?

Our reply:

Similar to 2-year recurrence, medium-chain acylcarnitines, acetylated polyamines, conjugated secondary bile acids and fatty acids were associated with death at 2 years. In contrast, ornithine, nicotinamide and tryptophan metabolites were higher in patients who were alive at 2 years (not shown). Regarding fecal taxa, *F. prausnitzii* strains were overrepresented in patients with favorable prognosis at T1 (not shown). Our clinician (Dr Bol) did not think that overall survival was meaningful in this paper. She wrote “I would suggest to delete this figure as well as all corresponding text as I believe 2 year survival data is too immature to draw any conclusions from for stage III melanoma patients. There are very little events (as we show in Table S2) as we would expect. Large studies might show 5 year overall survival data but even tend to wait for 10 year OS data”.

Reviewer #2 (Remarks to the Author): with expertise in microbiome, cancer

The report by Alves Costa Silva, C. et al aims to describe associations between fecal microbiome and blood metabolomic signatures with clinical outcomes in a randomized DC based clinical trial in patients with stage III melanoma. This report adds to the growing body of literature suggesting links between the composition of the gut microbiome and immune activation that aim to explain heterogenous responses to different types of cancer therapies including immunotherapy. The trial design, sample collection, and data generation presented herein offer numerous opportunities to evaluate associations between gut microbiomes and clinical outcomes; however, the results presented in this report are vastly limited by the analytical approach taken by the authors.

Our reply:

We warmly thank this referee for her/his careful reading and consideration.

Q1. The main limitation of this work is the lack of evaluation of the data in a longitudinal manner. T1 and T2 timepoints are presented in cross-sectional analyses but inaccurately labeled as longitudinal. The longitudinal sample collection of and evaluation of the stool and serum samples are a major highlight of this trial design, and the data should be evaluated in the same fashion. The authors reported differences in baselines between arms but failed to evaluate the data longitudinally within each arm to avoid the biases reported. Evaluating similarities and differences between groups at each timepoint is important, but given this is an interventional trial, evaluating pre vs. post within group and by response status in a paired manner and across groups will yield more meaningful results. This way, the authors could either use the baseline samples to evaluate predictor potential of response of baseline microbiome metrics as well as stability of microbiome over time comparing active arm vs. placebo. Moreover, the authors could derive stability measurements between T1 and T2 and determine if stability is associated with outcomes of interest.

Our reply:

This point is very well taken. We added longitudinal analysis for both omics to assess the impact of the DC on the evolution from T1 to T2, considering the clinical factors that could influence this dynamics (revised Figure S4, revised Figure S8, revised Figure 5), and a whole chapter for

the influence of this dynamic on the 2YR prediction in the result section entitled “Dynamic and integrative pathways” to address this issue as follows.

Using the XGBoost algorithm, coupled to a model explainer based on SHapley Additive exPlanations (SHAP) values for model interpretability (Référence: <https://dl.acm.org/doi/10.1145/2939672.2939785>), we corroborated that patients assigned to nDC treatment arm differed from PL patients as to primary BA and polyamines (Figure S7). Next, we re-analyzed the clinical relevance of all biological and clinical features to reduce dimension and allow prediction of 2Y-R taking into account their interactions and their slope of evolution overtime. We corroborated that BA (in particular chenodeoxycholic and cholic acid), FA and acetylcarnitines as well as a few pathways meaningful in other studies including tryptophan metabolism, vitamin B3 (trigonelline, 1-methylnicotinamide) and polyamines (acetylated metabolites of spermine/spermidine)) were predictors of 2Y-RFS at baseline (Figure 5A, AUC=0.732 (CI 95% : 0.660 - 0.804)). Next, the metabolite evolution between the two timepoints (T2-T1/T1) was added to the baseline value (T1) to identify if biomarker drifts could improve the prediction of the 2Y-RFS (model T2-T1/T1) and if treatment arms influenced this evolution (Figure S8A-C). The SHAP analysis indicated that the increase of FA (oleic acid) and acetylated polyamines and the decrease of ornithine (upstream of the polyamine pathway) and primary BA (glyco-cholic acid, chenodeoxycholic acid) were associated with an increased risk of 2Y-R (T2-T1/T1: AUC=0.72 (CI 95%: 0.66 - 0.78, Figure S8A). Even if the trajectory of BA and polyamines was associated with the prognosis of stage III melanoma, we could not conclude that nDC significantly alter their levels overtime (Figure S8B showing the most significant features among all SHAP parameters). We applied the same analysis integrating metagenomics and clinical features at T1 (Figure 5B), but obtaining better prediction at T2 for 2Y-RFS (Figure S8C) and confirming the prognostic impact of *F. prausnitzii* S15332. The circosplot highlights the anticorrelations inbetween these MGS and primary BA as well as positive correlations inbetween factors associated with dismal prognosis at T1 (Figure 5C). The multi-omics integrative model is depicted in Figure 5D.

Q2. All the p-values presented need to be corrected for multiple comparisons and statistical tests (instead of “model coefficient” report the test used) need to be mentioned alongside p values throughout the text and in figure legends.

Our reply:

The detailed list of p and q values for Figure 1 and Figure 3 MG analyses is now presented in Table S3 of the revised manuscript. Many features have a weak association with the treatment arm or the time point or 2Y survival and after FDR correction, are no longer significant, as stated in the text. This is likely due to relatively small sample size in each arm and time point, for the high number of MG+MB+clinical parameters, as well as the negative clinical outcome of the trial. We are aware that this work is an explanatory study from a small sample size relatively to the very high dimension of the data we analyzed, that cannot pass any strict statistical FDR correction. However, supported by the convergence of the standard statistical analysis and the machine learning approaches, and also by the literature, we are confident in the relevance of our findings to nail down clinically relevant results and draw hypothesis for a rationale-based prospective validation.

Q3. Metagenomic and metabolomic and other data reported in this manuscript should be deposited prior to final submission and accession numbers provided in the manuscript. Additionally, data can be made available through supplementary tables.

Our reply:

We have created two supplemental Tables detailing the raw statistical values of metagenomics (MG) and metabolomics (MB) for each time point and each treatment arm, for 2YR vs no-R (Tables S3A-C and S5A-B) in the revised version. MG and MB raw data are deposited under the accession numbers bioprojects PRJEB66197 for MG and Elsevier/Mendeley DOI: 10.17632/nzb653783h.1 for MB.

Q4. A full revision of all figures to check for y- and x-axes labels is required as well as figure legends. Labels should not be the sample size for each arm but rather descriptive titles for each group compared.

Our reply:

Sorry for this sloppiness. All panels have been double checked and appropriately labeled with a title.

Q5. I highly recommend having a statistician review and revise the manuscript and figures

Our reply:

The article was revised by a biostatistician (Dr Damien Drubay and mathematician Déborah Suissa) and data scientist experts in metagenomics (Prof. Nicola Segata) who co-sign this revised manuscript for their improvement by their input on the new statistical methods (SGD differential analysis, XGboost, and feature dynamic adjusted for clinical factors).

Minor Comments

- **Check for grammar, syntax,...**

Our reply:

We did our best to correct the article for grammar and syntax.

- **Words like “and”, “bacterium”, “spp.”, “model coefficient”, and numbers, including p values, should not be italicized.**

Our reply:

We reviewed the article and these terms are no longer italicized.

- **Avoid words and phrases like “some”, “similar to that”,**

Our reply:

We did our best to correct these sentences in the revised version.

Instead, or in addition of using “significantly correlated”, “a direct correlation”, report a p-value, test, and citation.

Our reply:

We changed sentences accordingly to include test, P-values, and citation.

In the abstract, I recommend rewording the statement below in the abstract to convey an accurate perspective of where the field is regards to using (fecal) microbiome composition as a predictor of response as a predictor for survival in melanoma. At best, we have observed associations between fecal microbiome composition and survival metrics which carry more power at a cohort-level than when multiple cohorts are combined. “The taxonomic composition of the gut microbiome earned its credentials among predictors of survival in melanoma, by influencing the peripheral and tumoral immune tonus.”

Our reply:

We mitigated this notion. Now, you can read: “The taxonomic composition of the gut microbiome earned its credentials among prognosis factors associated with survival in cancer, including melanoma patients, in part by influencing the peripheral and tumoral immune tonus.”

The sentences referring to Sipuleucel-T in prostate cancer in the introduction is not supporting any of the information presented thereafter. I suggest removing or addressing in the discussion in the context of the results presented.

Our reply:

We removed the sentences referring to Sipuleucel-T in prostate cancer in the introduction.

Here, we investigated.... could modulate “fecal” metagenomic and metabolomic profiles...

Our reply:

We changed the sentences accordingly: “Here, we investigated whether nDC loaded with TAA ex vivo could modulate fecal metagenomics (MG) and serum metabolomics (MB) profiles that might in turn, influence immunological and clinical outcomes.”

These sentences are not results and should be placed in the discussion: “In a meta-analysis incorporating new cohorts....1-year progression free survival rates. Bacteria associated ... and Blautia spp.”

“FA are involved in energy metabolism...and cancer cell metabolism. In tumor tissues, free FA...to feed into the TCA cycle”.

Our reply:

We took this advice in consideration in the revised version.

Avoid using terms such as “for the first time”. This is not the first-time shotgun metagenomic sequencing has been used to profiles stool samples from patients with late-stage melanoma.

Our reply:

In fact, to our knowledge, stool MG analyses have not been reported for (tumor free) stage III melanoma (in contrast to evidence accumulated for stage IV).

This is incorrect: beta diversity of fecal taxa distribution – please reword.

Our reply:

The sentence has been modified.

Citations missing for KEGG, MetaCyc, Bray-Curtis, MetaPhlAn4, HUMAnN3.6, Shannon diversity index, VEGAN, Wilcoxon test, Spearman test,...

Our reply:

We added the citations for KEGG, MetaCyc, Bray-Curtis, MetaPhlAn4, HUMAnN3.6, Shannon diversity index, VEGAN, Wilcoxon test and Spearman test, as well as for the new method added for this revision (XGBoost, Boruta, MICE, performance index optimism correction).

For the following statement: “Here, we infer from these findings... with melanoma recurrence”. Instead, present this inference as a hypothesis or a model in the discussion.

Our reply:

We mitigated the conclusions and did our best to rephrase this sentence. Indeed, this trial allowed to draw new working hypothesis for a rationale-based prospective validation.

• In the section “Taxonomic and metabolomic differences between treatment arms at randomization” – please reword the opening statement. Should not begin with “Hence” since it is the opening of a new results section.

Our reply:

The opening statement has been reworded.

Replace “monitored” with “assess with”, “evaluated by”, “determined by” in “(monitored by Bray-Curtis dissimilarity)”

Our reply:

We changed the article accordingly. “Monitored” was replaced by “assessed with”.

The Spearman test is used to determine if correlations exist, but it does not offer multi-omic data integration. Please check and reword statements.

Our reply:

We reworded the statement and created a specific section apostrophed “data integration” as detailed above.

• Box plots should show all data points to visualize the distribution of data like how it is presented in Figure 2D.

Our reply:

Boxplots have been changed to include all data points when possible.

Reviewer #3 (Remarks to the Author): with expertise in melanoma, immunotherapy, cancer

This translational science paper describes why the authors believe the microbiota is responsible for the negative results in a recent phase III clinical trial.

Q1. In general, the language of the text could be significantly shortened. For example, the first two pages of the results section could be made into a Table making it easier for the reader to follow.

Our reply:

We created a Table S4 to shorten the paragraph and the list of bacteria. However, the other referees asked for more details and this revised manuscript also aims at bringing some answers to their questions.

Q2. What type of supervised hierarchical clustering was utilized for the metabolomics experiments?

Our answer:

Data log₂ normalised / centered around the average abundance computed from all samples; Hierarchical clustering analysis using ward.D2 algorithm - Distance: Euclidean. Row: samples, columns: signals; red=higher than average, blue=lower than average, grey=missing;

Figure 2C is now detailing the following: Non supervised hierarchical clustering (Euclidean distance, ward linkage method) of the lipid metabolite abundances. The heatmap illustrates the changes in lipid metabolite abundance in the plasma of melanoma patients at T1 or T2, in each treatment arm for 2Y-R and noR outcomes. Significant metabolites were identified by Wilcoxon rank-sum test between patients with 2Y-R and noR outcomes are depicted in 2D.

Q3. This is an association study. What mechanism can the authors deduce from their studies? There will not be definitive experiments, but the patterns of MB and MG should be associated with T and DC function/biochemical processes (or even perhaps B cell function) associated with good responses in melanoma. This reviewer sees hints of this analysis, but it should be further developed.

Our reply:

In our revised manuscript, we created a result section integrating all the data and proposing landmark fingerprints associated with melanoma progression that need to be prospectively validated. You can now read the following that is depicted in new Figure 5 and new Figure S8 that are shown in this Point-by-point reply Figure 1 and 2 above.

“Dynamic and integrative pathways”

Using the XGBoost algorithm, coupled to a model explainer based on SHapley Additive exPlanations (SHAP) values for model interpretability (Référence: <https://dl.acm.org/doi/10.1145/2939672.2939785>), we corroborated that patients assigned to nDC treatment arm differed from PL patients as to primary BA and polyamines (Figure S7).

Next, we re-analyzed the clinical relevance of all biological and clinical features to reduce dimension and allow prediction of 2Y-R taking into account their interactions and their slope of evolution overtime. We corroborated that BA (in particular chenodeoxycholic and cholic acid), FA and acetylcarnitines as well as a few pathways meaningful in other studies including tryptophan metabolism, vitamin B3 (trigonelline, 1-methylnicotinamide) and polyamines

(acetylated metabolites of spermine/spermidine)) were predictors of 2Y-RFS at baseline (Figure 5A, AUC=0.732 (CI 95% : 0.660 - 0.804)). Next, the metabolite evolution between the two timepoints (T2-T1/T1) was added to the baseline value (T1) to identify if biomarker drifts could improve the prediction of the 2Y-RFS (model T2-T1/T1) and if treatment arms influenced this evolution (Figure S8A-C). The SHAP analysis indicated that the increase of FA (oleic acid) and acetylated polyamines and the decrease of ornithine (upstream of the polyamine pathway) and primary BA (glyco-cholic acid, chenodeoxycholic acid) were associated with an increased risk of 2Y-R (T2-T1/T1: AUC=0.72 (CI 95%: 0.66 - 0.78, Figure S8A). Even if the trajectory of BA and polyamines was associated with the prognosis of stage III melanoma, we could not conclude that nDC significantly alter their levels overtime (Figure S8B showing the most significant features among all SHAP parameters). We applied the same analysis integrating metagenomics and clinical features at T1 (Figure 5B), but obtaining better prediction at T2 for 2Y-RFS (Figure S8C) and confirming the prognostic impact of *F. prausnitzii* S15332. The circosplot highlights the anticorrelations inbetween these MGS and primary BA as well as positive correlations inbetween factors associated with dismal prognosis at T1 (Figure 5C). The multi-omics integrative model is depicted in Figure 5D (AUC=0.75, CI 95%: 0.68-0.80), where the MGS Ruminococcaceae bacterium SGB14899, *S. parasanguinis* SGB8071 and *F. prausnitzii* SGB15332 are consistently associated with better prognosis. Furthermore, the multi-omics integrative model corroborate MB signatures such as polyamines, FA (oleic acid), BA (chenodeoxycholic acid), acetylcarnitines and pathways including tryptophan metabolism (kynurenine), vitamin B3 (1-methylnicotinamide) are predictors of 2Y-RFS at T1.

Altogether, a number of biological serum soluble markers independent from clinical parameters impacted the survival of this cohort of stage III melanoma, that may not be directly inferred to the nDC treatment, including *F. prausnitzii* strains, fatty acids and acylcarnitines, biliary acids and polyamines, that interact together”.

In addition, immunomonitoring was performed in 39 patients included in the MIND-DC trial. Tumor antigen-specific and functional T cells were analyzed in blood using dextramer staining by flow cytometric analyses and in skin cultures of DTH- challenged sites (I. Jolanda M de Vries et al JCO 2005). HLA-A1- binding dextramers for gp100, tyrosinase, NY-ESO-1, MAGE-C2, MAGE-A3, HLA-A2- binding dextramers for gp100, tyrosinase, NY-ESO-1, MAGE-C2, MAGE-A3, and HLA-B35 binding dextramers for NY-ESO-1 and MAGE-A3 were used in 33, 33 and 15 patients respectively. Seven and four patients presented preexisting (T1) or developed (T2) dextramer- positive stainings respectively while 28 remained negative.

Correlative studies between the presence of detectable peptide vaccine-specific T cells in the blood with the MG-based taxonomic composition of stools at T1 revealed that vaccine-relevant cellular immunity was associated with health- and immunity-related bacteria (*F. prau. SGB15323*, *Blautia spp*, *Bifidobacterium catenulatum*, *Lachnospiraceae bacterium SGB4909*, *Lachnospiraceae bacterium SGB4909* and *Ruminococcus spp.*) (PBPR Figure 3A below). In addition, metabolic reads inferred from the MG gene alignment reinforced the potential role of the gut microbiome through the arginine and polyamine biosynthesis in this peripheral immune tonus post-vaccination (Figure S2B).

In parallel, DTH skin tests were performed between 1-2 weeks after the third injection of cycle 1. Depending on randomization, DC or PL were injected intradermally. After propagation in interleukin (rIL)-2, the recall responses of skin-test infiltrating lymphocytes (SKILs) to tumor and vaccine versus control peptides were monitored in ELISA for the detection of IFN γ , IL-2, IL-10 and IL-5. The number of melanoma patients exhibiting SKIL Th1 (IFN γ and IL-2) reactivity, SKIL Th2 (IL-10 or IL-5), or no vaccine/tumor peptide-specific cytokine release were 55, 7 and 37 respectively. Patients harboring intestinal *F. prausnitzii*, or *Butyricicoccus spp.* harbored mixed Th1/Th2 responses, while those carrying *Blautia spp.*, *Bifidobacterium catenulatum*, and *Ruminococcus spp.* exhibited Th2- geared DTH responses (PBPR Figure 3B). As expected from earlier studies in advanced lung cancer patients, *Eggerthellaceae unclass. spp* were associated with no recall responses.

Altogether, the fecal ecosystem may reflect the peripheral immune tonus of stage III melanoma patients with the presence of health-related *F. prausnitzii* associated with Th1/Th2 recall responses to tumor and vaccine peptides. These data have not been included in this paper because the last author contemplates to submit another paper detailing all the immune responses with various complementary methods.

Q4. Health-related microbiota other than. Recommend to make a table of the health related microbiota in addition to *F. prausnitzii*. If one did sub stratify the trial population in an exploratory analysis, does removing this confounder variable make a difference?

Our reply:

All Of Our Faecalibacterium prausnitzii may not be a confounder variable because the different strains do not reach the same effect.

Faecalibacterium prausnitzii is a very prevalent commensal species (>90% healthy individuals being positive). Indeed, its presence might be an indirect witness of a healthy status. Of note, five *F. prausnitzii* clades have been identified from 55 publicly available genomes and 92 high-quality metagenome assembled genomes. (iScience. 2022 Dec 22; 25(12): 105533). Using metaPhlan4, we identified 4 SGB (*SGB15316*, *SGB15318*, *SGB15322*, *SGB15323*). The prevalence and relative abundances of each strain differ between healthy volunteers and melanoma patients as well as between PL and DC (Figure S2C-D), specifically SGB15316 tended to be less prevalent and less abundant in melanoma (compared with HV) et in nDC (compared with PL) (Figure S2D). Two out of the four *F. prausnitzii* strains (SGB15316 and SGB15318) are clinically relevant when combined with Carnitine 12:0 to predict overall recurrence (novel Figure S8C-D). Other bacteria of potential relevance for the 2Y-R are detailed in Table S4 with p and q values (including the ones described for health-related status).

Q5. I Would recommend providing T and N staging given AJCC version 7.

Our reply:

We added the T and N staging given AJCC version 7 to Table S1 in the revised manuscript.

6. A pathway map would be expected synthesizing all the data to illustrate the biochemical principles and their relationship to DC and T cell biology. This should be a readable pathway map describing the major pathways and why the authors' hypotheses about microbiota may be correct. This reviewer understands there will not be definitive proof but more significant associations are needed.

Our reply:

We added a CIRCOSPLOT gathering all the clinical parameters, & metabolomics and metagenomics-based features that were statistically significant for predicting 2 year recurrence (with the AUC depicted in Figure S8C) in stage III melanoma at T1 and/or T2 in XGBoost with their respective correlations (shown in this point-by-point replay Figure 4 below, which is shown in Figure 5C of the revised manuscript).

The increase of FA (oleic acid) and acetylated polyamines (N8 acetylspermidine) and the decrease of ornithine (upstream of the polyamine pathway) and primary BA (glyco-cholic acid, chenodeoxycholic acid) were associated with an increased risk of 2YR (T2_minus_T1 & T1 : AUC=0.7190522 (CI 95% : 0.6566962 - 0.7751768, Figure S8A). The metabolite evolution between the two timepoints according to the treatment arm was assessed using linear regression adjusted for the trial arm, age, gender, tumor stage (III-b vs III-c), ECOG performance status, and BMI (details in the methods section), revealing very few and weak nDC-associated modulations (Figure S8B-C). Hence, in addition to fatty acids, biliary acids and polyamines are also important metabolites associated with the prognosis of stage III melanoma and that significant bias in their relative abundances could discriminate the nDC from the PL group at baseline, while DC did not significantly alter their levels overtime. We applied the same analysis integrating metagenomics and clinical features, obtaining better prediction of 2Y survival at T1 and confirming the prognostic impact of *F. prausnitzii* (Figure 5B of the revised manuscript). The circosplot highlights the anticorrelations between these MGS and the composition of BA and polyamines (Figure 5C of the revised manuscript and shown in the **PBPR Figure 4**). The integration of all omics and clinical parameters is now proposed in Figure

5D of the revised manuscript (PBPR Figure 1 above). The graphical scheme is illustrating in a simple way to this referee the overall scenario (PBPR Figure 5 below).

PBPR Figure 4= Figure 5C in the manuscript. Selection of the most significant features (clinics, metabolomics, metagenomics) predicting 2Y recurrence free-survival in stage III melanoma patients of the MIND-DC study and their correlations. Circosplot indicating correlations between common features described in the XGBoost machine learning model algorithm (ML). Features are clinical parameters, and MB +MG, monitored in serum at T1 or T2. SHAP values for each feature per patient are positive when the value of the feature increases the prediction of recurrence, negative otherwise, and are shown in Figure 5A-B of the revised manuscript. Each dot represents one patient and the color represents the value of each feature. Thickness of lines indicating an increasing positive (pink) or negative (blue) correlation.

[FIGURE REDACTED]

PBPR Figure 5. Graphical summary. Stage III melanoma patients may harbor aberrations in their lipid metabolism (carboxylic acids and middle chain acetylcarnitines) and biliary acid composition that correlated with deviated taxonomic composition of their stools (with relative under-representation of health-related bacteria such as distinct strains of *F. prausnitzii*). These perturbations were also associated with acetylated polyamines, and low levels of ornithine (and also abnormalities in tryptophan metabolites) that may accelerate recurrence of the disease after surgery, despite adjuvant vaccination with autologous dendritic cells.

REVIEWER COMMENTS

Reviewer #1 (Remarks to the Author):

The manuscript improved considerably, and all my points were addressed adequately. I don't have further concerns.

Reviewer #2 (Remarks to the Author):

Main considerations:

The report will benefit from a careful selection of results to be presented in main text and figures to convey results in clear and concise way. Most of the analyses presented are almost raw outputs of statistical evaluations without any consideration of biological and/or statistical significance. This distracts from the main points the authors are trying to make.

Other points.

- taxonomic and metabolomic features with low, near 0 coefficients should be removed from main figures and corrected p values should be added. For instance, all taxonomic features presented in Fig 1C are within -0.04 and +0.04. Similar cases in Fig 1D and Fig 3B.
- prevalence of features analyzed should be represented together with the differential abundance/correlation analysis as these methods are sensitive to uneven distribution of features within groups.
- exact p values and statistical tests should be reported for every analysis.
-

Reviewer #3 (Remarks to the Author):

Thank you for the rebuttal document.

The only comment is that I would just add that certain pieces of information were not collected.

"Sadly, dietary habits and life stylerelated pieces of information were not collected at that time (back 8 years ago when conceiving the study). So, we conclude there was no overt clinical bias in the randomization process."

Villejuif-Grand-Paris,
November 13th, 2023

REVIEWER COMMENTS

Reviewer #1 (Remarks to the Author):

The manuscript improved considerably, and all my points were addressed adequately. I don't have further concerns.

Our response:

We thank the reviewer for her/his comments.

Reviewer #2 (Remarks to the Author):

Main considerations:

The report will benefit from a careful selection of results to be presented in main text and figures to convey results in clear and concise way. Most of the analyses presented are almost raw outputs of statistical evaluations without any consideration of biological and/or statistical significance. This distracts from the main points the authors are trying to make.

Our response:

The article has been written focusing on the main messages, converging to the circosplot depicted at the end highlighting the main points (Fig. 5B), that metabolomics and to some extent, metagenomics can be useful tools to identify dismal prognosis in stage III melanoma, either at baseline or shortly after the first vaccination. Eleven of the sixteen figures have been modified to give consideration to the most impactful conclusions.

P1: Taxonomic and metabolomic features with low, near 0 coefficients should be removed from main figures and corrected p values should be added. For instance, all taxonomic features presented in Fig 1C are within -0.04 and +0.04. Similar cases in Fig 1D and Fig 3B.

Our response:

We agree with this point to select the significant and most biologically relevant associations in metagenomics analyses. We did not introduce a threshold on the model coefficient to remove “near 0 coefficients”, since all of them are weak ($P < 0.05$ and none with $Q < 0.2$) but point to the potential significance of *Faecalibacterium prausnitzii*. Although none of them did pass the FDR correction, albeit passing the nominal $P < 0.05$ using Arcsin-sqrt as well as Centered-log-ratio, we used the second independent method (the XGBoost algorithm, coupled to a model explainer based on SHapley Additive exPlanations (SHAP) values for model interpretability) already unveiled in the previous version that corroborated independent and significant prognosis parameters (such as *F. prausnitzii*, Figure 5A) that significantly separated the placebo from the nDC group at T2 ($p < 0.03$) (Figure S2B, left panel). Of note, Figure 6C highlights the clinical relevance of *F. prausnitzii* combined with middle chain carnitines to identify the subgroup of stage III melanoma patients endowed with a favorable outcome within the whole population. As recommended by this reviewer, we paid attention to remove the lowest variables selected in this Boruta XGBoost +SHAP model (Figure 5A).

P2: Prevalence of features analyzed should be represented together with the differential abundance/correlation analysis as these methods are sensitive to uneven distribution of features within groups.

Our response:

We agree with this view. The prevalence of the metagenomics features are represented in tables S3A-C.

P3: exact p values and statistical tests should be reported for every analysis.

Our response:

We carefully reviewed the article and the exact p values and statistical tests are reported for every analysis (either in figures legends and/or in tables).

Reviewer #3 (Remarks to the Author):

Thank you for the rebuttal document.

The only comment is that I would just add that certain pieces of information were not collected. "Sadly, dietary habits and life style related pieces of information were not collected at that time (back 8 years ago when conceiving the study). So, we conclude there was no overt clinical bias in the randomization process."

Our response:

We thank the reviewer for her/his comments. This comment “Of note, dietary habits and life style -related pieces of information were not collected at the time of protocol conception, restraining some important correlations with metagenomics and metabolomics results” has been added the second revised version of our manuscript in material and methods (section “trial design”).

REVIEWERS' COMMENTS

Reviewer #2 (Remarks to the Author):

While the revised manuscript represents a substantial improvement over its previous version, several areas still require attention and correction. Firstly, the title suggests the involvement of gut microbiome species in immune system reprogramming. While this is a prevailing hypothesis in the field, the authors did not present conclusive evidence beyond associations, some of which are only borderline significant, with quantified metabolites and imputed metabolic functions. The manuscript lacks experimental evidence necessary to support the claim of immune system reprogramming. I recommend revising the title to accurately reflect the analysis and results obtained.

In terms of result analysis and presentation, especially for instances involving differential abundance analyses, it would be beneficial to include a prevalence plot illustrating the presence and abundance of each evaluated feature. This would provide readers with a comprehensive overview of the underlying data.

The presence of *Streptococcus salivarius* and *Streptococcus parasanguinis* in feces, known oral cavity colonizers, raises questions about whether this is a result of oral seeding. I suggest including a discussion on the biological relevance of oral bacteria colonizing the gut and its potential impact on systemic immunity within the context of this trial.

In reference to line 194, there seems to be a misuse of the term "evolution." It appears that the authors intend to convey a shift or change in the composition of the fecal microbiome. Clarifying this language would enhance the accuracy of the manuscript.

Concerning Figure 2, Panel A, there is difficulty distinguishing T1, T2, 2Y-R, and 2Y-noR. Enhancing the clarity of this figure, perhaps through color differentiation or labeling, would improve the reader's ability to interpret the data.

Rather than using terms like "major differences" and "massive shifts," it is recommended to provide include statistical measures and directionality to support these claims. Proper

statistical evaluation is crucial to substantiate statements about differences and shifts in the data.

Villejuif-Grand-Paris,
December 29th, 2023

REVIEWER COMMENTS

Reviewer #2 (Remarks to the Author):

While the revised manuscript represents a substantial improvement over its previous version, several areas still require attention and correction. Firstly, the title suggests the involvement of gut microbiome species in immune system reprogramming. While this is a prevailing hypothesis in the field, the authors did not present conclusive evidence beyond associations, some of which are only borderline significant, with quantified metabolites and imputed metabolic functions. The manuscript lacks experimental evidence necessary to support the claim of immune system reprogramming. I recommend revising the title to accurately reflect the analysis and results obtained.

Our response:

We thank the reviewer for her/his comments. The title refer to metabolic but not immune system reprogramming and has been changed to “Influence of microbiota-associated metabolic reprogramming on clinical outcome in patients with melanoma from the randomized adjuvant dendritic cell-based MIND-DC trial” following editorial revision.

In terms of result analysis and presentation, especially for instances involving differential abundance analyses, it would be beneficial to include a prevalence plot illustrating the presence and abundance of each evaluated feature. This would provide readers with a comprehensive overview of the underlying data.

Our response:

We agree with this view. The prevalence of each evaluated metagenomics feature is represented in tables S3A-C.

The presence of *Streptococcus salivarius* and *Streptococcus parasanguinis* in feces, known oral cavity colonizers, raises questions about whether this is a result of oral seeding. I suggest including a discussion on the biological relevance of oral bacteria colonizing the gut and its potential impact on systemic immunity within the context of this trial.

Our response:

It is indeed an interesting discussion and we thank the reviewer for her/his suggestion. The article discussion has not been modified to focus on the main points and stick to editorial guidelines (the main text should be limited to 5,000 words).

In reference to line 194, there seems to be a misuse of the term "evolution." It appears that the authors intend to convey a shift or change in the composition of the fecal microbiome. Clarifying this language would enhance the accuracy of the manuscript.

Our response:

We thank the reviewer for her/his comments. We clarified this language (the term "evolution" has been replaced by "shift") would enhance the accuracy of the manuscript.

Concerning Figure 2, Panel A, there is difficulty distinguishing T1, T2, 2Y-R, and 2Y-noR. Enhancing the clarity of this figure, perhaps through color differentiation or labeling, would improve the reader's ability to interpret the data.

Our response:

We thank the reviewer for her/his comments. Both color coding and labelling has been reviewed to follow editorial guidelines.

Rather than using terms like "major differences" and "massive shifts," it is recommended to provide include statistical measures and directionality to support these claims. Proper statistical evaluation is crucial to substantiate statements about differences and shifts in the data.

Our response:

We carefully reviewed the article to avoid those terms and the exact p values and statistical tests are reported through the article.